# Zero-shot causal learning

**Hamed Nilforoshan**[*1]    **Michael Moor**[*1]    **Yusuf Roohani**[2]    **Yining Chen**[1]
**Anja Šurina**[3]    **Michihiro Yasunaga**[1]    **Sara Oblak**[4]    **Jure Leskovec**[1]

[1]Department of Computer Science, Stanford University
[2]Department of Biomedical Data Science, Stanford University
[3]School of Computer and Communication Sciences, EPFL
[4]Department of Computer Science, University of Ljubljana

Correspondence to: hamedn@cs.stanford.edu, mdmoor@cs.stanford.edu, jure@cs.stanford.edu

## Abstract

Predicting how different interventions will causally affect a specific individual is important in a variety of domains such as personalized medicine, public policy, and online marketing. There are a large number of methods to predict the effect of an existing intervention based on historical data from individuals who received it. However, in many settings it is important to predict the effects of novel interventions (e.g., a newly invented drug), which these methods do not address. Here, we consider zero-shot causal learning: predicting the personalized effects of a novel intervention. We propose CaML, a causal meta-learning framework which formulates the personalized prediction of each intervention's effect as a task. CaML trains a single meta-model across thousands of tasks, each constructed by sampling an intervention, its recipients, and its nonrecipients. By leveraging both intervention information (e.g., a drug's attributes) and individual features (e.g., a patient's history), CaML is able to predict the personalized effects of novel interventions that do not exist at the time of training. Experimental results on real world datasets in large-scale medical claims and cell-line perturbations demonstrate the effectiveness of our approach. Most strikingly, CaML's zero-shot predictions outperform even strong baselines trained directly on data from the test interventions.

## 1    Introduction

Personalized predictions about how an intervention will causally affect a specific individual are important across many high impact applications in the physical, life, and social sciences. For instance, consider a doctor deciding whether or not to prescribe a drug to a patient. Depending on the patient, the same drug could either (a) cure the disease, (b) have no effect, or (c) elicit a life-threatening adverse reaction. Predicting which effect the drug will have for each patient could revolutionize healthcare by enabling personalized treatments for each patient.

The causal inference literature formalizes this problem as conditional average treatment effects (CATE) estimation, in which the goal is to predict the effect of an intervention, conditioned on patient characteristics ($X$). When natural experiment data is available, consisting of individuals who already did and did not receive an intervention, a variety of CATE estimators exist to accomplish this task [1, 3, 16, 23, 30, 34, 44, 55, 36, 70]. These methods can then predict the effect of an *existing* intervention ($W$) on a new individual ($X'$).

However, in many real-world applications natural experiment data is entirely unavailable, and yet CATE estimation is critical. For instance, when new drugs are discovered, or new government policies

---

[*]Equal contribution. Code is available at: `https://github.com/snap-stanford/caml/`

37th Conference on Neural Information Processing Systems (NeurIPS 2023).

are passed, it is important to know the effect of these novel interventions on individuals and subgroups in advance, i.e., before anybody is treated. There is thus a need for methods that can predict the effect of a *novel* intervention ($W'$) on a new individual ($X'$) in a zero-shot fashion, i.e., without relying on *any* historical data from individuals who received the intervention.

Generalizing to novel interventions is especially challenging because it requires generalizing across two dimensions simultaneously: to new interventions and new individuals. This entails efficiently "aligning" newly observed interventions to the ones previously observed in the training data.

**Present work.** Here, we first formulate the zero-shot CATE estimation problem. We then propose CaML (**Ca**usal **M**eta-**l**earning), a general framework for training a single meta-model to estimate CATE across many interventions, including novel interventions that did not exist at the time of model training (Figure 1). Our key insight is to frame CATE estimation for each intervention as a separate meta-learning task. For each task observed during training, we sample a retrospective natural experiment consisting of both (a) individuals who did receive the intervention, and (b) individuals who did not receive the intervention. This natural experiment data is used to estimate the effect of the intervention for each individual (using any off-the-shelf CATE estimator), which serves as the training target for the task.

In order to achieve zero-shot generalization to new interventions, we include information ($W$) about the intervention (e.g., a drug's attributes), in the task. We then train a single meta-model which fuses intervention information with individual-level features ($X$) to predict the intervention's effect. Our approach allows us to predict the causal effect of novel interventions, i.e., interventions without sample-level training data, such as a newly discovered drug (Figure 1). We refer to this capability as *zero-shot causal learning*.

In our experiments, we evaluate our method on two real-world datasets—breaking convention with the CATE methods literature which typically relies on synthetic and semi-synthetic datasets. Our experiments show that CaML is both scalable and effective, including the application to a large-scale medical dataset featuring tens of millions of patients. Most strikingly, CaML's zero-shot performance exceeds even strong baselines that were trained directly on data from the test interventions. We further discover that CaML is capable of zero-shot generalization even under challenging conditions: when trained only on single interventions, at inference time it can accurately predict the effect of combinations of novel interventions. Finally, we explain these findings, by proving a zero-shot generalization bound.

## 2 Related work

We discuss recent work which is most closely related to zero-shot causal learning, and provide an extended discussion of other related work in Appendix B. Most CATE estimators do not address novel interventions, requiring that all considered interventions be observed during training. A notable exception is recent methods which estimate CATE for an intervention using structured information about its attributes [25, 35]. In principle, these methods can also be used for zero-shot predictions. These methods estimate CATE directly from the raw triplets ($W, X, Y$), without considering natural experiments, by tailoring specific existing CATE estimators (the S-learner [44] and Robinson decomposition [55], respectively) to structured treatments. The main drawback of these approaches is that they are inflexible, i.e., they are restricted to using a single estimator and are unable to take advantage of the recent advances in the broader CATE estimation literature (e.g., recently developed binary treatment estimators [16, 21, 40]). This is a limitation because any single CATE estimator can be unstable across different settings [15]. Notably, the estimators which these methods build on have already been shown to result in high bias in many domains [44, 37, 10, 16]. Likewise, we find that these methods struggle with zero-shot predictions (Section 6). CaML's key difference from prior work is that we construct a separate task for each training intervention by synthesizing natural experiments. This allows us to (a) flexibly wrap any existing CATE estimator to obtain labels for each task, and thus take advantage of the most recent CATE estimation methods and (b) leverage meta-learning, which requires task-structured data. Consequently, CaML is able to achieve strong zero-shot performance (Section 6).

## 3 Background: single-intervention CATE estimation

Each task in the CaML framework consists of estimating conditional average treatment effects (CATEs) for a single binary treatment. In this section, we first provide background on CATE

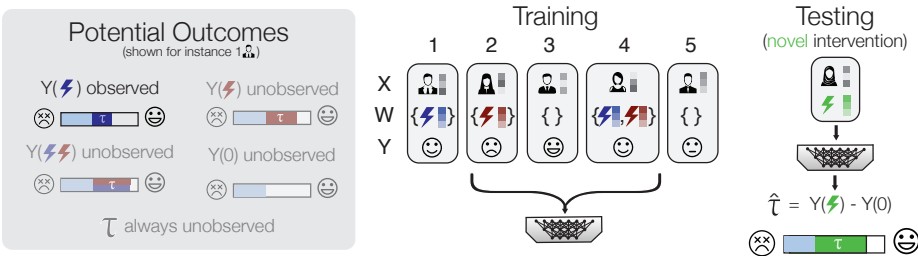

Figure 1: Overview of the zero-shot causal learning problem. Each individual has features $(X)$, an intervention with features $(W)$, and an outcome $(Y)$. Lightning bolts (⚡) represent interventions (*e.g.* drugs). The personalized effect of an intervention $(\tau)$ is always unobserved. The goal is to predict the $\tau$ for a novel intervention $(W')$ and individual $(X')$ that did not exist during training.

estimation under this simple case of a single treatment $(W)$ and outcome $(Y)$, and subsequently generalize it to our zero-shot setting. Under a single intervention and outcome, we consider $n$ independent observations $P_1, \ldots, P_n$ drawn from a distribution $\mathcal{P}$. For unit $i = 1, ..., n$, $P_i = (W_i, X_i, Y_i) \sim \mathcal{P}$ collects: a binary or continuous outcome of interest $Y_i \in \mathcal{Y} \subset \mathbb{R}$, instance features (i.e., pre-treatment covariates) $X_i \in \mathcal{X} \subset \mathbb{R}^d$, and a treatment-assignment indicator $W_i \in \{0, 1\}$. We use the Neyman-Rubin potential outcomes framework [33], in which $Y_i(1), Y_i(0)$ reflect the outcome of interest either under treatment $(W_i = 1)$, or under control $(W_i = 0)$, respectively. In our running medical example, $Y_i(1)$ is the health status if exposed to the drug, and $Y_i(0)$ is the health status if not exposed to the drug. Notably, the *fundamental problem of causal inference* is that we only observe one of the two potential outcomes, as $Y_i = W_i \cdot Y_i(1) + (1 - W_i) \cdot Y_i(0)$ (e.g., either health status with or without drug exposure can be observed for a specific individual, depending on whether they are prescribed the drug). However, it is possible to make personalized decisions by estimating treatment effects that are tailored to the attributes of individuals (based on features $X$). Thus, we focus on estimating $\tau(x)$, known as the conditional average treatment effect (CATE):

$$\text{CATE} = \tau(x) = \mathbb{E}_{\mathcal{P}}\Big[Y(1) - Y(0) \mid X = x\Big] \tag{1}$$

A variety of methods have been developed to estimate $\tau(x)$ from observational data [16]. These rely on standard assumptions of unconfoundedness, consistency, and overlap [52]. *Unconfoundedness*: there are no unobserved confounders, i.e. $Y_i(0), Y_i(1) \perp\!\!\!\perp W_i \mid X_i$. *Consistency*: $Y_i = Y_i(W_i)$, i.e. treatment assignment determines whether $Y_i(1)$ or $Y_i(0)$ is observed. *Overlap*: Treatment assignment is nondeterministic, such that for all $x$ in support of $X$: $0 < P(W_i = 1 \mid X_i = x) < 1$.

## 4 Zero-shot causal learning

In many real-world settings (*e.g.* drugs, online A/B tests) novel interventions are frequently introduced, for which no natural experiment data are available. These settings require zero-shot CATE estimates. The zero-shot CATE estimation problem extends the prior section, except the intervention variable $W_i$ is no longer binary, but rather contains rich information about the intervention: $W_i \in \mathcal{W} \subset \mathbb{R}^e$ (e.g., a drug's chemistry), where $W_i = 0$ corresponds to a sample that did not receive any intervention. Thus, each intervention value $w$ has its own CATE function that we seek to estimate:

$$\text{CATE}_w = \tau_w(x) = \mathbb{E}_{\mathcal{P}}\Big[Y(w) - Y(0) \mid X = x\Big], \tag{2}$$

During training, we observe $n$ independent observations $P_1, \ldots, P_n$ drawn from a distribution $\mathcal{P}$. Each $P_i = (W_i, X_i, Y_i) \sim \mathcal{P}$. Let $\mathcal{W}_{seen}$ be set of all interventions observed during training. The zero-shot CATE estimation task consists of estimating CATE for a novel intervention that was never observed during training:

*Problem* 1 (Zero-shot CATE estimation). **Given** $n$ training observations $(W_1, X_1, Y_1), \ldots, (W_n, X_n, Y_n)$ drawn from $\mathcal{P}$ containing intervention information, individual features, and outcomes... **estimate** $\tau_{w'}(x)$ for a novel intervention $w' \notin \mathcal{W}_{seen}$.

This problem formulation extends in a straightforward manner to combinations of interventions, by allowing a single intervention $W_i$ to consist of a set of intervention vectors. CaML supports combinations of interventions, as we elaborate on in Section 4.1

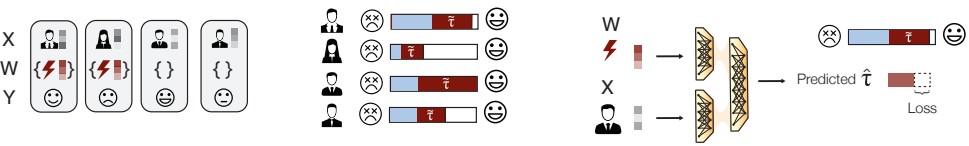

Figure 2: Visual illustration of the CaML (causal meta-learning) framework. (1) We sample a task (i.e., an intervention) and a natural experiment from the training data consisting of individuals who either received the intervention (W={⚡}), or did not (W={}). Each individual has features ($X$) and an outcome ($Y$), and the intervention also has information ($W$) (e.g., a drug's attributes). (2) For each individual we estimate the effect of the intervention on the outcome (pseudo-outcomes $\tilde{\tau}$). (3) We predict an individual's pseudo-outcomes $\tilde{\tau}$ using a model that fuses $X$ and $W$. CaML is trained by repeating this procedure across many tasks and corresponding natural experiments.

**CaML overview.** We propose a novel framework for estimating CATE across multiple interventions, even including ones that were never encountered during training. Our framework consists of three key components (Figure 2). First, we formulate CATE estimation as a meta-learning problem in which each task corresponds to the CATE estimation for a unique intervention. A task dataset for a given intervention is constructed by sampling a natural experiment of all individuals who received the intervention, and a sample of individuals who did not. Tasks are augmented with intervention information ($W$). Synthesizing these natural experiments allows us to compute a noisy CATE label $\tilde{\tau}$ using any off-the-shelf estimator ($\tilde{\tau}$ is referred to as pseudo-outcomes by the causal inference literature [16]). Finally, we train a single meta-model to predict these labels using individual-level ($X$) and intervention-level ($W$) information, such that it is able to generalize to novel tasks, i.e., estimating CATE for novel interventions.

The CaML framework incorporates three important design considerations: *(1) Single meta-model*. In domains such as electronic health records and online marketing, we observe that large-scale datasets contain thousands of interventions with rich feature information ($W$). Instead of training a separate model for each intervention, CaML trains a single meta-model that can estimate CATE across all interventions. This approach lets us leverage shared structure across tasks and generalize to novel interventions that were not present during training. *(2) Pseudo-outcomes*. Instead of directly modeling the response surfaces $\mathbb{E}[Y(w) \mid X = x]$ and $\mathbb{E}[Y(0) \mid X = x]$, we use pseudo-outcomes for each intervention to train our model. This approach is informed by recent studies indicating bias in estimating CATE from direct predictions of observed outcomes [10, 44]. CaML outperforms strong baselines that meta-learn $Y(w)$ and $Y(0)$ directly, as demonstrated in our experiments (see Tables 2 and 3, rows S-learner and T-learner with meta-learning). *(3) Discrete tasks from continuous interventions*. CaML takes advantage of the extensive literature on CATE estimation for single, binary interventions. By creating a natural experiment for each intervention, CaML taps into this literature and benefits from the high performance of recently developed nonparametric CATE estimators [16, 55, 44].

CaML identifies CATE for novel interventions under the assumptions that: (1) for each observed intervention $w$, $\tau_w(x)$ is identifiable under the binary treatment assumptions (unconfoundedness, consistency, and overlap) in Section 3. This allows for valid training labels for each task. (2) $\tau_w(x) = \tau(w, x)$, i.e., a global function $\tau(w, x)$ unifies all intervention-specific CATE functions, (3) $\tau(w, x)$ is continuous in $w$. This allows the model to smoothly extrapolate the treatment effect to new interventions that are close to observed interventions in the intervention space. Lastly, (4) $W$ follows a continuous distribution.

## 4.1 Meta-dataset

We formulate CATE estimation as a meta-learning problem. For this, each task refers to CATE estimation for a distinct intervention. Interventions as well as tasks in our meta-dataset are jointly indexed by $j \in \mathbb{N}$ with $1 \leq j \leq K$, such that we can refer to the $j$-th intervention information with $w^{(j)}$.

We then construct a meta-dataset $D$ in the following way:

$$D = \left\{ \left( D_{\text{treated}}^{(j)} \cup D_{\text{control}}^{(j)}, w^{(j)} \right) \right\}_{j=1}^{K}, \quad \text{with} \tag{3}$$

$$D_{\text{treated}}^{(j)} = \{(X_i, Y_i) \mid W_i = w^{(j)}\} \text{ and } D_{\text{control}}^{(j)} = \{(X_i, Y_i) \mid W_i = 0)\}. \tag{4}$$

$D^{(j)}$ denotes the natural experiment dataset for task $j$, composed of a treated group (instances which received the intervention, i.e. $W_i = w^{(j)}$) and control group (instances which did not receive any intervention, i.e. $W_i = 0$). Each sample $i$ represents an individual, for which the quantities $(X_i, Y_i)$ are collected as introduced in Section 3. In practice, we down-sample both groups (i.e. to 1 million samples for the treated and control groups) in our large-scale experiments.

We augment each task dataset $D^{(j)}$ with intervention information, $w^{(j)} \in \mathbb{R}^e$, for zero-shot generalization to new interventions [35, 18, 87, 39]. The form of $w^{(j)}$ varies with the problem domain — for text interventions, it could be a language model's text embedding [79, 84, 58], while biomedical treatments can be represented as nodes in a knowledge graph [8, 49]. Additionally, domain-specific features, like treatment categories from an ontology, may be included in $w^{(j)}$. To handle combinations of interventions (e.g., pairs of drugs), we aggregate the $w$ for each intervention using an order-invariant pooling operation (we used the sum operator), and sample a separate natural experiment for individuals who received the full combination.

## 4.2 Estimating pseudo-outcomes

We next estimate the training targets for each task (i.e. intervention) in the meta-dataset. The training target ($\tilde{\tau}^{(j)}$) is an unbiased, but noisy, estimate of CATE. More formally, for each task $j$ (which points to the natural experiment dataset for intervention $w^{(j)}$), we estimate $\tilde{\tau}^{(j)}$, where $\mathbb{E}_{\mathcal{P}}[\tilde{\tau}^{(j)}|X = x] = \tau_{w^{(j)}}(x)$. Thus, $\tilde{\tau}_i^{(j)}$ denotes the target for the $i$-th sample in the $j$-th task (indexing will be omitted when it is clear from context). We refer to these targets as pseudo-outcomes, following prior literature [16]. For prior work on pseudo-outcomes, refer to Appendix B. In Appendix E we demonstrate why these pseudo-outcomes provide an unbiased training objective. For a detailed explanation on the necessity of using pseudo-outcomes instead of directly modeling $Y(w)$ and $Y(0)$, please see [44, 16, 10].

CaML is agnostic to the specific choice of pseudo-outcome estimator. Thus, we assume a function $\eta(D^{(j)})$ which takes as input a task dataset $D^{(j)} \in D$ and returns a vector containing the pseudo-outcomes $\tilde{\tau}$ for each sample in the task. We extend each task dataset $D^{(j)}$ with the pseudo-outcomes, such that a sample holds the elements $(X_i, Y_i, \tilde{\tau}_i)$. Our key insight is that by collecting these pseudo-outcomes across multiple tasks, and predicting them using a combination of intervention and individual information $(W, X)$ we can develop a CATE estimator which generalizes to novel interventions. In practice, we use the RA-learner [17] and treat pseudo-outcome estimation as a data pre-processing step (Appendix C.6).

## 4.3 Meta-model training

Given $m$ target outcomes $Y_1, ..., Y_m$ (e.g., different drug side effects), our goal is then to learn a model $\Psi_\theta : \mathbb{R}^e \times \mathbb{R}^d \to \mathbb{R}^m$ that for parameters $\theta$ minimizes

$$\theta^* = \underset{\theta}{\arg\min} \; \mathbb{E}_{j \sim U(D)} \; \mathbb{E}_{W,X,\tilde{\tau} \sim D^{(j)}} \left[ L\left(\Psi_\theta\right) \right], \tag{5}$$

where $U(D)$ denotes the discrete uniform distribution over the tasks of the meta-dataset $D$, and where $L(f)$ refers to a standard loss function between the pseudo-outcomes and the model output, i.e., $L(f) = (\tilde{\tau} - f(w,x))^2$. To assess whether the model generalizes to novel tasks, we partition our meta-dataset by task, into non-overlapping subsets $D = D_{\text{train}} \cup D_{\text{val}} \cup D_{\text{test}}$. During training, $\Psi_\theta$ is optimized on training tasks $D_{\text{train}}$. We validate and test this model on $D_{\text{val}}$ and $D_{\text{test}}$, which are thus unseen during training tasks. While the CaML framework is agnostic to a specific training strategy, we based our approach (Algorithm 1) on the Reptile meta-learning algorithm [53] which we find performs better compared to straightforward empirical risk minimization (c.f. Section 6). For this, the objective is slightly modified to

$$\theta^* = \underset{\theta}{\arg\min} \; \mathbb{E}_{j \sim U(D)} \left[ L\left(A_{D^j}^k\left(\Psi_\theta\right)\right) \right], \tag{6}$$

**Algorithm 1** The CaML algorithm

**Require:** meta-dataset $D$, meta-model $\Psi_\theta$ with initialized parameters $\theta$, hyperparameter $k$.
  **for** iteration $= 1, 2, \ldots, L$ **do**
    $j \leftarrow$ SAMPLETASK()
    $D_{\text{treat}}^{(j)}, D_{\text{ctrl}}^{(j)}, w^{(j)} \leftarrow$ QUERYTASKDATA($j$)
    $\tilde{\tau}^{(j)} \leftarrow$ ESTIMATEPSEUDOOUTCOMES($D_{\text{treat}}^{(j)}, D_{\text{ctrl}}^{(j)}$)
    $\theta' \leftarrow$ ADAPT($(D_{\text{treat}}^{(j)}, D_{\text{ctrl}}^{(j)}), \tilde{\tau}^{(j)}, w^{(j)}, \Psi_\theta, k$)
    $g \leftarrow \theta - \theta'$ {Reptile gradient}
    $\theta \leftarrow \theta - \beta g$ {Gradient step for meta-model $\Psi_\theta$}
  **end for**
  **return** $\Psi_\theta$

**function** ADAPT(Data $D$, Pseudo-outcomes $\tilde{\tau}$, Intervention information $w$, Model $\Psi_\theta$, # of Steps $k$)
  $\Psi_\theta' \leftarrow$ Create copy of $\Psi_\theta$
  **for** s $= 1, 2, \ldots, k$ **do**
    Draw batch of size $b$ from $D$.
    Compute loss $\mathcal{L}_s$ by feeding instances through model, conditioned on task:
    $\mathcal{L}_s = \frac{1}{b} \sum_{i=1}^{b} (\tilde{\tau}_i - \Psi_\theta'(w_i, x_i))^2$
    Update parameters of $\Psi_\theta'$:
    $\theta \leftarrow \theta - \alpha \nabla \mathcal{L}_s$
  **end for**
**end function**

where $A_D^k \colon \mathcal{F} \to \mathcal{F}$ represents the operator that updates a model $f \in \mathcal{F}$ using data sampled from the dataset $D$ for $k$ gradient steps. This operator is defined in more detail as the ADAPT routine in Algorithm 1. Note that depending on the choice of CATE estimator, this routine iterates only over treated samples of a task dataset $D^{(j)}$ (as in our experiments), or over all samples, including untreated ones.

## 4.4 CaML architecture

To parameterize $\Psi_\theta$, we propose a simple but effective model architecture (see Section 6):

$$\Psi_\theta(w, x) = \text{MLP}_1([\tilde{w}; \tilde{x}]), \text{ with } \tilde{x} = \text{MLP}_2(x) \text{ and } \tilde{w} = \text{MLP}_3(w), \tag{7}$$

where $[\cdot\,;\cdot]$ denotes concatenation. Equation 7 shows that the intervention information $w$ and individual features $x$ are encoded separately into dense vectors $\tilde{w}$ and $\tilde{x}$, respectively. Our MLPs consist of layers of the form $g(z) = z + \text{ReLU}(\text{Linear}(z))$.

## 5 Theoretical analysis

We now consider zero-shot causal learning from a theoretical perspective. Under simplified assumptions, we bound the prediction error in the zero-shot setting.

We formulate the setting as a supervised learning problem with noisy labels (pseudo-outcomes) where we learn a smooth function $f = \Psi(w, x) \to \tau$ among a family $\mathcal{F}$. We focus on $\tau \in \mathbb{R}$, and assume $\tau \in [0, 1]$ without loss of generality, since we can normalize $\tau$ to this range. The training dataset has $n$ interventions with $m$ samples each, i.e. first $n$ *i.i.d.* draws from $P_W$: $w^{(1)}, \ldots, w^{(n)}$ and then for each $w^{(j)}$, $m$ *i.i.d.* draws from $P_X$: $x_1^{(j)}, \ldots, x_m^{(j)}$.

The main theorem quantifies the rate that combining information across different interventions helps with zero-shot performance. We prove a finite-sample generalization bound for the ERM variant of CaML. The ERM is a special case of ADAPT with $k = 1$ that is more conducive to rigorous analysis. The advantage of Reptile over ERM is orthogonal and we refer the readers to the original discussion [54]. We assume the estimated pseudo-outcomes $\tilde{\tau}$ during training satisfy $\tilde{\tau} = \tau + \xi$ where $\xi$ is an independent zero-mean noise with $|\xi| \leq \epsilon$ almost surely for some $\epsilon \geq 0$,

$$\hat{f} = \min_{f \in \mathcal{F}} \hat{L}(f) = \min_{f} \frac{1}{nm} \sum_{j=1}^{n} \sum_{i=1}^{m} (f(w^{(j)}, x_i^{(j)}) - \tilde{\tau}_i^{(j)})^2.$$

The test error is $L(f) = \mathbb{E}_{W, X, \tau}[(f(w, x) - \tau)^2]$. Let $f^* = \min_f L(f)$. We bound the excess loss $L(\hat{f}) - L(f^*)$. Our key assumption is that interventions with similar features $W$ have similar effects in expectation. We assume that all functions in our family are smooth with respect to $W$, i.e., $\forall f \in \mathcal{F}, \mathbb{E}_{W, X}\left[\|\partial f/\partial W\|_2^2\right] \leq \beta^2$.

**Theorem 1.** *Under our assumptions, with probability $1 - \delta$,*

$$L(\hat{f}) \leq L(f^*) + 8(1+\epsilon)R_{nm}(\mathcal{F}) + 8\sqrt{\frac{(1+\epsilon)R_{nm}(\mathcal{F})\log(1/\delta)}{n}} + \frac{2\log(1/\delta)}{3n} +$$

$$(1+\epsilon)\sqrt{\frac{(32C\beta^2 + 2(1+\epsilon)^2/m)\log(1/\delta)}{n}}$$

where $R_{nm}$ is a novel notion of zero-shot Rademacher complexity defined in equation (9); $C$ is a Poincaré constant that only depends on the distribution of $W$. For large $n, m$, the leading terms are the function complexity $R_{nm}(\mathcal{F})$, and an $O(\sqrt{1/n})$ term with a numerator that scales with $\beta$ and $(1+\epsilon)^2/m$. This validates our intuition that when the intervention information $W$ is more informative of the true treatment effects (smaller $\beta$), and when the estimation of $\tau$ in the training dataset is more accurate, the performance is better on novel interventions. Please refer to Section A for the full proof. Compared to standard generalization bound which usually has a $\sqrt{1/n}$ term, our main technical innovation involves bounding the variance by the smoothness of the function class plus Poincaré-type inequalities. When $\beta$ is much smaller than 1 we achieve a tighter bound.

| Dataset | Samples | Features ($X$) | Outcome ($Y$) | Intervention type | Intervention information ($W$) |
|---|---|---|---|---|---|
| Claims | Patients | Patient history (binned counts of medical codes) | Pancytopenia onset | Drug intake (prescription) | Drug embedding (knowledge graph) |
| LINCS | Cell lines | Cancer cell encyclopedia | Expression of landmark genes (DEG) | Perturbagen (small molecule) | Molecular embeddings (RDKit) |

Table 1: High-level overview of our two experimental settings. Details in Appendix C.1.

## 6 Experiments

We explore to what extent zero-shot generalization is practical when predicting the effects of interventions. We thus design two novel evaluation settings using real-world data in domains where zero-shot CATE estimation will be highly impactful: (1) Health Insurance Claims: predicting the effect of a drug on a patient, and (2) LINCS: predicting the effect of a perturbation on a cell. We use new datasets because existing causal inference benchmarks [31, 73] focus on a single intervention. By contrast, zero-shot causal learning must be conceptualized in a multi-intervention setting.

**Zero-shot Evaluation**. Each task corresponds to estimating CATE for a single intervention, across many individual samples (e.g. patients). We split all tasks into meta-training/meta-validation, and a hold-out meta-testing set for evaluating zero-shot predictions (Table 2, unseen drugs for Claims and Table 3, unseen molecular perturbations in LINCS). For the Claims dataset, we also consider the challenging setting of combinations of unseen drugs (Table 5).

Each meta-validation and meta-testing task contains a natural experiment of many samples (e.g., patients) who received the unseen intervention, and many control samples who did not receive the intervention. The same patient (Claims) or cell-line (LINCS) can appear in multiple tasks (if they received different interventions at different times). Thus, to ensure a fair zero-shot evaluation, we exclude all samples who have ever received a meta-testing intervention from meta-val/meta-train. Similarly, we exclude all meta-validation patients from meta-train. Details on holdout selection are provided in Appendix C.2.

Table 1 gives an overview of both benchmarks. In the Claims dataset, we compare zero-shot predictions with strong single-intervention baselines which cannot generalize to unseen interventions. To do so, we further split each task in meta-validation and meta-testing into a train/test (50/50) split of samples. These baselines are trained on a task's train split, and all methods are evaluated on the test split of the meta-testing tasks. On the LINCS dataset, as each task consists of $< 100$ cells, single-intervention baselines performed weakly and are excluded from analysis.

**Baselines.** We compare the zero-shot performance of CaML to two distinct categories of baselines. (1) *Trained directly on test interventions*. These are strong CATE estimators from prior work and

can only be trained on a single intervention. Thus, we train a single model on each meta-testing task's train split, and evaluate performance on its test split. This category includes T-learner [44], X-learner [44], RA-learner [16], R-learner [55], DragonNet [72], TARNet [70], and FlexTENet [17].

(2) *Zero-shot* baselines are trained across all meta-training tasks and are able to incorporate intervention information ($W$). These methods are thus, in principle, capable of generalizing to unseen interventions. We use GraphITE [25] and Structured Intervention Networks (SIN) [35]. We also introduce two strong baselines which learn to directly estimate $Y(w)$ and $Y(0)$ by meta-learning across all training interventions, without using pseudo-outcomes: S-learner and T-learner with meta-learning. These extend the S-learner and T-learner from prior work [44] to incorporate intervention information ($W$) in their predictions. We elaborate on implementation details of baselines in Appendix C.7. For details on hyperparameter search and fair comparison, see Appendix C.1.

**Ablations.** In our first ablation experiment (w/o meta-learning), we trained the CaML model without meta-learning, instead using the standard empirical risk minimization (ERM) technique [78]. Our second ablation (w/o RA-learner) assesses the sensitivity of CaML's performance to different pseudo-outcome estimation strategies. For further details on how these ablation studies were implemented, see Appendix C.3. We discuss the key findings from these ablations in Section 6.3.

## 6.1 Setting 1: Personalized drug side effect prediction from large-scale medical claims

Our first setting (Claims) is to predict the increased likelihood of a life-threatening side effect caused by a drug prescription. We leverage a large-scale insurance claims dataset of over 3.5 billion claims across 30.6 million patients in the United States[2]. Each datestamped insurance claim contains a set of diagnoses (ICD-10 codes), drug prescriptions (DrugBank ID), procedures (ICD-10 codes), and laboratory results (LOINC codes). Laboratory results were categorized by whether the result was high, low, normal, abnormal (for non-continuous labs), or unknown.

Interventions are administration of one drug ($n = 745$), or two drugs ($n = 22{,}883$) prescribed in combination. Time of intervention corresponds to the *first* day of exposure. Intervention information ($W$) was generated from pre-trained drug embeddings from a large-scale biomedical knowledge graph [8] (Appendix C). We compute drug combination embeddings as the sum of the embeddings of the constituent drugs. We focus on the binary outcome ($Y$) of the occurrence of the side effect pancytopenia within 90 days of intervention exposure. Pancytopenia is a deficiency across all three blood cell lines (red blood cells, white blood cells, and platelets). Pancytopenia is life-threatening, with a 10-20% mortality rate [38, 43], and is a rare side effect of many common medications [42] (*e.g.* arthritis and cancer drugs), which in turn require intensive monitoring of the blood work. Following prior work [24], patient medical history features ($X$) were constructed by time-binned counts of each unique medical code (diagnosis, procedure, lab result, drug prescription) at seven different time scales before the drug was prescribed, resulting in a total of 443,940 features. For more details, refer to Appendix C.1.

**Metrics** We rely on best practices for evaluating CATE estimators in observational data, as established by recent work [86, 11], which recommend to assess treatment rules by comparing subgroups across different quantiles of estimated CATE. We follow the high vs. others RATE (rank-weighted average treatment effect) approach from Yadlowsky et. al [86], which computes the difference in average treatment effect (ATE) of the top $u$ percent of individuals (ranked by predicted CATE), versus all individuals (for more details, see Appendix C.1). For instance, RATE @ 0.99 is the difference between the top 1% of the samples (by estimated CATE) vs. the average treatment effect (ATE) across all samples, which we would expect to be high if the CATE estimator is accurate. Note that estimates of RATE can be negative if model predictions are inversely associated with CATE. We elaborate on the RATE computation in Appendix C.1.

The real-world use case of our model is preventing drug prescription for a small subset of high-risk individuals. Thus, more specifically, for each task $j$, intervention $w_j$ in the meta-dataset, and meta-model $\Psi_\theta$, we compute $RATE @ u$ for each $u$ in $[0.999, 0.998, 0.995, 0.99]$ across individuals who received the intervention. We use a narrow range for $u$ because pancytopenia is a very rare event occurring in less than 0.3% of the patients in our dataset. Hence, in a real-world deployment scenario, it is necessary to isolate the small subset of high-risk patients from the vast majority of patients for whom there is no risk of pancytopenia onset.

---

[2]Insurance company undisclosed per data use agreement.

|  | RATE @$u$ ($\uparrow$) | | | | Recall @$u$ ($\uparrow$) | | | | Precision @$u$ ($\uparrow$) | | | |
|---|---|---|---|---|---|---|---|---|---|---|---|---|
|  | 0.999 | .998 | 0.995 | 0.99 | 0.999 | 0.998 | 0.995 | 0.99 | 0.999 | 0.998 | 0.995 | 0.99 |
| Random | 0.00 | 0.00 | 0.00 | 0.00 | 0.00 | 0.00 | 0.01 | 0.00 | 0.00 | 0.00 | 0.00 | 0.00 |
| T-learner | 0.32 | 0.26 | 0.16 | 0.10 | 0.12 | 0.18 | 0.26 | 0.31 | 0.36 | 0.29 | 0.18 | 0.11 |
| X-learner | 0.06 | 0.05 | 0.04 | 0.03 | 0.02 | 0.04 | 0.08 | 0.12 | 0.09 | 0.07 | 0.06 | 0.05 |
| R-learner | 0.19 | 0.17 | 0.12 | 0.08 | 0.06 | 0.1 | 0.19 | 0.26 | 0.24 | 0.21 | 0.15 | 0.11 |
| RA-learner | 0.47 | 0.37 | 0.23 | 0.14 | 0.17 | 0.26 | 0.38 | 0.45 | 0.54 | 0.42 | 0.26 | 0.16 |
| DragonNet | 0.09 | 0.07 | 0.05 | 0.04 | 0.03 | 0.05 | 0.08 | 0.11 | 0.15 | 0.12 | 0.08 | 0.06 |
| TARNet | 0.15 | 0.12 | 0.07 | 0.05 | 0.05 | 0.08 | 0.12 | 0.14 | 0.18 | 0.15 | 0.09 | 0.06 |
| FlexTENet | 0.10 | 0.09 | 0.06 | 0.04 | 0.04 | 0.06 | 0.11 | 0.16 | 0.15 | 0.13 | 0.09 | 0.06 |
| GraphITE | 0.19 | 0.12 | 0.05 | 0.03 | 0.07 | 0.08 | 0.09 | 0.10 | 0.23 | 0.14 | 0.07 | 0.04 |
| SIN | 0.00 | 0.00 | 0.00 | 0.00 | 0.00 | 0.00 | 0.01 | 0.02 | 0.01 | 0.01 | 0.01 | 0.01 |
| S-learner w/ meta-learning | 0.21 | 0.16 | 0.09 | 0.05 | 0.08 | 0.11 | 0.15 | 0.16 | 0.25 | 0.18 | 0.1 | 0.06 |
| T-learner w/ meta-learning | 0.40 | 0.31 | 0.18 | 0.11 | 0.15 | 0.22 | 0.32 | 0.38 | 0.45 | 0.35 | 0.21 | 0.13 |
| CaML - w/o meta-learning | 0.39 | 0.31 | 0.18 | 0.11 | 0.15 | 0.22 | 0.32 | 0.39 | 0.45 | 0.35 | 0.22 | 0.14 |
| CaML - w/o RA-learner | 0.45 | 0.36 | 0.22 | **0.14** | 0.16 | 0.24 | 0.34 | 0.41 | 0.48 | 0.38 | 0.26 | **0.16** |
| CaML (ours) | **0.48** | **0.38** | **0.23** | 0.13 | **0.18** | **0.27** | **0.38** | **0.45** | **0.54** | **0.43** | **0.26** | 0.16 |

**Trained on test intervention**

(Best underlined)

**Zero-shot**

(Best in bold)

Table 2: Performance results for the Claims dataset (predicting the effect of drug exposure on pancytopenia onset from patient medical history). Key findings are (1) CaML outperforms all zero-shot baselines (RATE is 18-27% higher than T-Learner w/ meta-learning, the strongest zero-shot baseline) (2) CaML performs stronger (up to 8× higher RATE values) than 6 of the 7 baselines which are trained directly on the test interventions, and performs comparably to the strongest baseline trained directly on the test interventions (RA-learner). Mean is reported across all runs; standard deviations included in (Appendix Table 4). Analogous trends hold for generalization to *pairs* of unseen drugs (Appendix Table 5).

Additionally, because our meta-testing dataset consists of individuals treated with drugs known to cause pancytopenia, observational metrics of recall and precision are also a rough *proxy* for successful CATE estimation (and highly correlated to RATE, Table 2). Thus, as secondary metrics, we also compute $Recall @ u$ and $Precision @ u$ for the same set of thresholds as RATE, where a positive label is defined as occurrence of pancytopenia after intervention.

## 6.2 Setting 2: Cellular gene expression response due to perturbation

Our second setting (LINCS) is to predict how a cell's gene expression ($Y$) will respond to intervention from perturbagen (small molecule compound such as a drug). This is a critical problem as accurately predicting intervention response will accelerate drug-discovery. We use data for 10,325 different perturbagens from the LINCS Program [74]. Each perturbagen corresponds to a different small molecule. Molecular embeddings were generated using the RDKit featurizer [46] and used as intervention information ($W$). Outcomes ($Y$) of interest are post-intervention gene expression across the top-50 and top-20 differentially expressed landmark genes (DEGs) in the LINCS dataset. We did not look at all 978 genes since most do not show significant variation upon perturbation. We use 19,221 features ($X$) from the Cancer Cell Line Encyclopedia (CCLE) [22] to characterize each cell-line ($n = 99$), each of which correspond to unperturbed gene expression measured in a different lab environment using a different experimental assay. For more details, see Appendix C.1.

**Metrics**. A key advantage of experiments on cells is that at evaluation time we can observe both $Y(0)$ and $Y(1)$ for the same cell line $X$, through multiple experiments on clones of the same cell-line in controlled lab conditions. In the LINCS dataset, $Y(0)$ is also measured for all cells which received an intervention. Thus, we can directly compute the precision in estimating heterogeneous effects (PEHE) on all treated cells in our meta-testing dataset, an established measure for CATE estimation performance analogous to mean-squared error [30] (see Appendix C.1).

## 6.3 Key findings

*CaML's zero-shot predictions outperform baselines with direct access to the target intervention.* In the medical claims setting, single intervention baselines (Tables 2, dark grey rows) are the highest performing baselines as we train them directly on the meta-test intervention. Still, CaML outperforms 6 out of 7 of these baselines (up to 8× higher RATE) and achieves comparable performance to the strongest of these baselines, the RA-learner. Furthermore, CaML strongly outperforms alternative zero-shot CATE estimators (RATE is 18-27% higher than T-Learner w/ meta-learning, the strongest zero-shot baseline). In the LINCS data, multi-intervention learners are strongest as

|  | PEHE 50 DEGs ($\downarrow$) | PEHE 20 DEGs ($\downarrow$) |
| --- | --- | --- |
| Mean. | 3.78 | 4.11 |
| GraphITE | 3.58 $\pm$ 0.023 | 3.82 $\pm$ 0.011 |
| SIN | 3.78 $\pm$ 0.001 | 4.06 $\pm$ 0.001 |
| S-learner w/ meta-learning | 3.63 $\pm$ 0.004 | 3.90 $\pm$ 0.004 |
| T-learner w/ meta-learning | 3.61 $\pm$ 0.007 | 3.85 $\pm$ 0.006 |
| CaML - w/o meta-learning | 3.57 $\pm$ 0.006 | 3.79 $\pm$ 0.004 |
| CaML - w/o RA-learner | 4.28 $\pm$ 0.517 | 4.60 $\pm$ 0.413 |
| CaML (ours) | **3.56** $\pm$ 0.001 | **3.78** $\pm$ 0.005 |

Table 3: Performance results for the LINCS dataset (predicting the effect of an unseen perturbation on the gene expression of an unseen cell-line). CaML outperforms all baselines. Improvement is largest for the 20 most differentially expressed genes, where most signal is expected.

there are only a small number of instances (cell lines) per intervention[3]. CaML outperforms both single-intervention and multi-intervention learners by drawing from both of their strengths—it allows us to use strong CATE estimation methods (i.e. the RA-learner) which previously were restricted to single interventions, while sharing information across multiple interventions.

*CaML learns to generalize from single interventions to combinations of unseen interventions (drug pairs).* We evaluate CaML's performance in the challenging setting of predicting the personalized effects of combinations of two drugs which are both unseen during training, while only training on interventions consisting of single drugs. CaML achieves strong performance results (see Appendix Table 5), surpassing the best baseline trained on the test tasks, and outperforms all zero-shot baselines, across all 12 metrics.

*Understanding CaML's performance results.* Our ablation studies explain that CaML's performance gains are due to (1) our meta-learning formulation and algorithm (in contrast to the w/o meta-learning row, in which ERM is used to train the model), and (2) the flexible CATE estimation strategy, allowing to take advantage of recently developed CATE estimators previously restricted to single interventions (in contrast to the w/o RA-learner row, in which an alternative pseudo-outcome estimator is used). Lastly, (3) comparison to existing binary intervention CATE estimators trained separately on each meta-testing intervention (Table 2, grey rows) shows that we gain from learning from thousands interventions. See Appendix C.3 for details on ablations.

## 7  Conclusion

We introduce a novel approach to predict the effects of novel interventions. CaML consistently outperforms state-of-the-art baselines, by unlocking zero-shot capability for many recently developed CATE estimation methods which were previously restricted to studying single interventions in isolation. While our study is limited to retrospective data, we plan to prospectively validate our findings. Future work includes designing new model architectures and CATE estimators which learn well under the CaML framework, developing new metrics to evaluate zero-shot CATE estimators, as well as more generally exploring novel learning strategies that enable zero-shot causal learning.

**Societal impacts**. In high-stakes decision-making inaccurate predictions can lead to severe consequences. It is important not to overly rely on model predictions and proactively involve domain experts, such as doctors, in the decision-making process. Additionally, it is crucial to ensure that underserved communities are not disadvantaged by errors in treatment effect estimates due to underrepresentation in the training data. Important avenues for achieving equitable CATE estimation in future work include process-oriented approaches (i.e., evaluating model errors for underserved demographics), and outcome-oriented methods (i.e., gauging model impacts on demographic utility) [12, 57, 69, 2]. Furthermore, the deployment of CATE models could raise privacy concerns. These models typically require access to individual patient data to estimate personalized treatment effects accurately. Ensuring the privacy and security of this sensitive information is crucial to avoid potential data breaches or unauthorized access, which could harm patients and erode public trust in healthcare systems.

---

[3]Single-task baselines excluded from Table 3: all performed similar or worse than mean baseline due to low task sample size.

## Acknowledgements

We are deeply grateful to Stefan Wager for his invaluable insights and extensive contributions to our discussions. We thank Emma Pierson, Kexin Huang, Kaidi Cao, Yanay Rosen, Johann Gaebler, Maria Brbić, Kefang Dong, June Vuong for helpful conversations. H.N was supported by a Stanford Knight-Hennessy Scholarship and the National Science Foundation Graduate Research Fellowship under Grant No. DGE-1656518. M.M. was supported by DARPA N660011924033 (MCS), NIH NINDS R61 NS11865, GSK, Wu Tsai Neurosciences Institute. A.S and S.O were supported by the American Slovenian Education Foundation (ASEF) fellowship. M.Y was supported by the Microsoft Research PhD fellowship. Y.R was supported by funding from GlaxoSmithKline LLC. Y.C. was supported by Stanford Graduate Fellowship and NSF IIS 2045685. We also gratefully acknowledge the support of Stanford HAI for Google Cloud Credits, DARPA under Nos. HR00112190039 (TAMI), N660011924033 (MCS); ARO under Nos. W911NF-16-1-0342 (MURI), W911NF-16-1-0171 (DURIP); NSF under Nos. OAC-1835598 (CINES), OAC-1934578 (HDR), CCF-1918940 (Expeditions), NIH under No. 3U54HG010426-04S1 (HuBMAP), Stanford Data Science Initiative, Wu Tsai Neurosciences Institute, Amazon, Docomo, GSK, Hitachi, Intel, JPMorgan Chase, Juniper Networks, KDDI, NEC, and Toshiba.

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

# A  Zero-shot Rademacher complexity and Proof of Theorem 1

## A.1  Problem setup and assumptions

Let $w \in \mathcal{W} \subseteq \mathbb{R}^e$ denote an intervention and $x \in \mathcal{X} \subseteq \mathbb{R}^d$ denote an individual that received it. Assume the outcome to predict is a scalar $y \in [0, 1]$. The hypothesis class is $\mathcal{F} = \{f : (w, x) \to y\}$. The dataset has $n$ interventions with $m$ independent units which received each intervention, i.e. first $n$ *i.i.d.* draws from $P_W$ and then $m$ *i.i.d.* draws from $P_X$ for each $w^{(j)}$. During training we have access to noisy estimate $\tilde{y} = y + \xi$ where $\xi$ is an independent noise with $\mathbb{E}\xi = 0$ and $|\xi| \leq \epsilon$ almost surely. We are tested directly on $y$.

The ERM is

$$\hat{f} = \min_f \hat{L}(f) = \min_f \frac{1}{nm} \sum_{j=1}^n \sum_{i=1}^m (f(w^{(j)}, x_i^{(j)}) - \tilde{y}_i^{(j)})^2.$$

The test error is

$$L(f) = \mathbb{E}_{w,x,y}(f(w, x) - y)^2 \tag{8}$$

and let $f^* = \min_f L(f)$.

We are interested in bounding the excess error $L(\hat{f}) - L(f^*)$.

Our key assumption is that interventions with similar attributes ($w$) have similar effects in expectation. More concretely, we assume that all hypotheses in our family are smooth with respect to $w$:

**Assumption 2.**

$$\forall f \in \mathcal{F}, \mathbb{E}_{w,x}\left[\left\|\frac{\partial f}{\partial w}\right\|_2^2\right] \leq \beta^2.$$

Furthermore, we assume that $P_W$ satisfies a Poincaré-type inequality:

**Assumption 3.**  For some constant $C$ that only depends on $P_W$, for any smooth function $F$,

$$Var_w[F(w)] \leq C\mathbb{E}\left[\|\nabla_w F(w)\|_2^2\right].$$

For example, $P_W$ can be any of the following distributions:

- Multivariate Gaussian: $w \in \mathbb{R}^e \sim \mathcal{N}(\mu, \Sigma)$ for some vector $\mu \in \mathbb{R}^e$ and positive semi-definite matrix $\Sigma \in \mathbb{R}^{e \times e}$;

- $w \in \mathbb{R}^e$ has independent coordinates; each coordinate has the symmetric exponential distribution $1/2e^{-|t|}$ for $t \in \mathbb{R}$.

- $P_W$ is a mixture over base distributions satisfying Poincaré inequalities, and their pair-wise chi-squared distances are bounded.

- $P_W$ is a mixture of isotropic Gaussians in $\mathbb{R}^e$.

- $P_W$ is the uniform distribution over $\mathcal{W} \subset \mathbb{R}^e$, which is open, connected, and bounded with Lipschitz boundary.

We note that isotropic Gaussian can approximate any smooth densities in $\mathbb{R}^e$ [41] (since RBF kernels are universal), showing that Assumption 3 is fairly general. We define a novel notion of function complexity specialized to the zero-shot setting. Intuitively, it measure how well we can fit random labels, which is first drawing $n$ interventions and $m$ recipients for each intervention. For examples of concrete upper bound on zero-shot Rademacher complexity see section A.4.

$$R_{nm}(F) = \frac{1}{nm}\mathbb{E}_{w,x,\sigma} \sup_f \sum_{j=1}^n \sum_{i=1}^m \sigma_i^j f(w^{(j)}, x_i^{(j)}) \tag{9}$$

where $\sigma_i^j$ are independently randomly drawn from $\{-1, 1\}$.

## A.2 Formal theorem statement

**Theorem 4.** *Under Assumptions 2 3, with probability $1 - \delta$,*

$$L(\hat{f}) \leq L(f^*) + 8(1+\epsilon)R_{nm}(\mathcal{F}) + 8\sqrt{\frac{(1+\epsilon)R_{nm}(\mathcal{F})\log(1/\delta)}{n}}$$

$$+(1+\epsilon)\sqrt{\frac{(32C\beta^2 + \frac{2(1+\epsilon)^2}{m})\log(1/\delta)}{n}} + \frac{2\log(1/\delta)}{3n}.$$

## A.3 Proof of the main theorem

We define the population loss on the noisy label $\widetilde{L}(f) = \mathbb{E}_{w,x,\tilde{y}}\left(f(w,x) - \tilde{y}\right)^2$. Due to independence of $\xi$, $\mathbb{E}_{w,x,y,\xi}(f(w,x) - y - \xi)^2 = \mathbb{E}_{w,x,y}(f(w,x) - y)^2 + \mathbb{E}[\xi^2] = L(f) + \mathbb{E}[\xi^2]$ for any $f$, so $L(\hat{f}) - L(f^*) = \widetilde{L}(\hat{f}) - \widetilde{L}(f^*)$. We shall focus on bounding the latter.

We first need a lemma that bounds the supremum of an empirical process indexed by a bounded function class.

**Lemma 5** (Theorem 2.3 of [7]). *Assume that $X_j$ are identically distributed according to $P$, $\mathcal{G}$ is a countable set of functions from $\mathcal{X}$ to $\mathbb{R}$ and, and all $g \in \mathcal{G}$ are $P$-measurable, square-integrable, and satisfy $\mathbb{E}[g] = 0$. Suppose $\sup_{g \in \mathcal{G}} \|g\|_\infty \leq 1$, and we denote $Z = \sup_g \left|\sum_{j=1}^n g(X_j)\right|$. Suppose $\sigma^2 \geq \sup_{g \in \mathcal{G}} Var(g(X_j))$ almost surely, the for all $t \geq 0$, we have*

$$\Pr\left[Z \geq \mathbb{E}Z + \sqrt{2t(n\sigma^2 + 2\mathbb{E}Z)} + \frac{t}{3}\right] \leq e^{-t}.$$

We apply Lemma 5 with $X_j = (w^{(j)}, x_1^j, \ldots, x_m^j, \tilde{y}_1^j, \ldots, \tilde{y}_m^j)$, $g(X_j) = \left(\frac{1}{m}\sum_i(f(w^{(j)}, x_i^{(j)}) - \tilde{y}_i^{(j)})^2 - \widetilde{L}(f)\right)$, $\sigma^2 = \sup_{f \in \mathcal{F}}(Var(\frac{1}{m}\sum_i(f(w^{(j)}, x_i^{(j)}) - \tilde{y}_i^{(j)})^2))$, $t = \log(1/\delta)$. Since $f - \tilde{y} \in [-1, 1]$, $g \in [-1, 1]$. With probability $1 - \delta$,

$$n\sup_f\left|\widehat{L}(f) - \widetilde{L}(f)\right| \leq n\mathbb{E}\sup_f\left|\widehat{L}(f) - \widetilde{L}(f)\right| + \sqrt{2\log\frac{1}{\delta}\left(n\sigma^2 + 2n\mathbb{E}\sup_f\left|\widehat{L}(f) - \widetilde{L}(f)\right|\right)} + \frac{1}{3}\log\frac{1}{\delta}.$$

Multiplying both sides by $1/n$, and using $\sqrt{a+b} \leq \sqrt{a} + \sqrt{b}$,

$$\sup_f\left|\widehat{L}(f) - \widetilde{L}(f)\right| \leq \mathbb{E}\sup_f\left|\widehat{L}(f) - \widetilde{L}(f)\right| + 2\sqrt{\frac{\mathbb{E}\sup_f\left|\widehat{L}(f) - \widetilde{L}(f)\right|\log(1/\delta)}{n}} + \sqrt{\frac{2\sigma^2\log(1/\delta)}{n}} + \frac{\log(1/\delta)}{3n}. \tag{10}$$

The next lemma bounds the variance $\sigma^2$ in equation (10).

**Lemma 6.**

$$\forall f \in \mathcal{F}, Var_{w^{(j)}, x_{1\ldots m}^j, \tilde{y}_{1\ldots m}^j}\left[\frac{1}{m}\sum_{i=1}^m(f(w^{(j)}, x_i^{(j)}) - \tilde{y}_i^{(j)})^2\right] \leq 4(1+\epsilon)^2 C\beta^2 + \frac{(1+\epsilon)^4}{4m}.$$

*Proof of Lemma 6.* Using the law of total variance, if we write

$$g(w^{(j)}, x_{1\ldots m}^j, \tilde{y}_{1\ldots m}^j) = \frac{1}{m}\sum_{i=1}^m(f(w^{(j)}, x_i^{(j)}) - \tilde{y}_i^{(j)})^2,$$

then

$$Var[g] = Var_w\left[\mathbb{E}_{x,\tilde{y}}[g(w, x, \tilde{y}) \mid w]\right] + \mathbb{E}_h\left[Var_{x,\tilde{y}}[g(w, x, \tilde{y}) \mid w]\right] \tag{11}$$

To bound the first term of equation (11), we use Poincaré-type inequalities in Assumption 3. For each of the example distributions, we show that they indeed satisfy Assumption 3.

**Lemma 7.** *Each of the example distributions in Assumption 3 satisfies a Poincare-type inequality.*

*Proof.*        • When $P_W$ is the uniform distribution over $\mathcal{W} \in \mathbb{R}^e$, which is open, connected, and bounded with Lipschitz boundary, we use Poincaré–Wirtinger inequality [60] on the smooth function $\mathbb{E}[g \mid w]$: For some constant $C$ that only depends on $P_W$,

$$Var_w[\mathbb{E}[g \mid w]] \leq C\mathbb{E}\left[\|\nabla_w \mathbb{E}[g \mid w]\|_2^2\right]. \tag{12}$$

$C$ is the Poincaré constant for the domain $\mathcal{W}$ in $L_2$ norm. It can be bounded by $1/\lambda_1$ where $\lambda_1$ is the first eigenvalue of the negative Laplacian of the manifold $\mathcal{W}$ [88]. Many previous works study the optimal Poincaré constants for various domains [45]. For example, when $w$ is uniform over $\mathcal{W}$ which is a bounded, convex, Lipschitz domain with diameter $d$, $C \leq d/\pi$ [59].

We can apply probabilistic Poincaré inequalities over non-Lebesgue measure $P_W$:

• When $w \sim \mathcal{N}(\mu, \Sigma)$, we use the Gaussian Poincaré inequality (see *e.g.* Theorem 3.20 of [6] and using change of variables),

$$Var[F(w)] \leq \mathbb{E}[\langle \Sigma \nabla_w F(w), \nabla_w F(w) \rangle].$$

We apply this with $F(w) = \mathbb{E}[g \mid w]$. Since $\mathbb{E}[v^\top A v] = \mathbb{E}[Tr(v^\top A v)] = \mathbb{E}[Tr(Avv^\top)] = Tr(A\mathbb{E}[vv^\top]) \leq \|A\|_2 \mathbb{E}\left[\|v\|_2^2\right]$,

$$Var_w[\mathbb{E}[g \mid w]] \leq \|\Sigma\|_2 \mathbb{E}\left[\|\nabla_w \mathbb{E}[g \mid w]\|_2^2\right],$$

which satisfies equation (12) with $C = \|\Sigma\|_2$.

• When $w \in \mathbb{R}^e$ has independent coordinates $w_1, \ldots, w_e$ and each coordinate has the symmetric exponential distribution $1/2e^{-|t|}$ for $t \in \mathbb{R}$, we first bound a single dimension using Lemma 4.1 of [47], which says for any function $k \in L^1$,

$$Var(k(w_i)) \leq 4\mathbb{E}\left[k'(w_i)^2\right]$$

which, combined with the Efro-Stein inequality (Theorem 3.1 of [6]),

$$Var(F(w)) = \mathbb{E}\sum_{i=1}^{e} Var(F(w) \mid w_1, \ldots, w_{i-1}, w_{i+1}, \ldots, w_n),$$

yields:

$$Var(F(w)) \leq 4\mathbb{E}\left[\|F'(w)\|_2^2\right]$$

which satisfies equation (12) with $C = 4$.

Lastly, we consider the case where $P_W$ is a mixture over base distributions satisfying Poincaré inequalities. We first consider the case where the pair-wise chi-squared distances are bounded. Next, we show that mixture of isotropic Gaussians satisfies Poincaré inequality without further condition on pair-wise chi-squared distances.

• When $\{P_W^q\}_{q \in \mathcal{Q}}$ is a family of distributions, each satisfying Poincaré inequality with constant $C^q$, and $P_W$ is any mixture over $\{P_W^q\}_{q \in \mathcal{Q}}$ with density $\mu$, let $K_P(\mu) = ess_\mu \sup_q C^q$, which is an upper bound on the base Poincaré constants almost surely, and $K_{\chi^2}^P(\mu) = \mathbb{E}_{q,q' \sim \mu}[(1 + \chi^2(P_W^q \| P_W^{q'}))^p]^{1/p}$, which is an upper bound on the pairwise $\chi^2$-divergence. Using Theorem 1 of [9] we get that $P_W$ satisfies Poincaré inequality with constant $C$ such that $C \leq K_P(\mu)(p^* + K_{\chi^2}^P(\mu))$ where $p^*$ is the dual exponent of $p$ satisfying $1/p + 1/p^* = 1$.

As an example, when base distributions are from the same exponential family and the natural parameter space is affine, such as mixture of Poisson or Multinomial distributions, the pair-wise chi-squared distances are bounded (under some additional conditions) and hence the mixture satisfies Poincaré inequality. More formally, let $p_\theta(x) =$

$\exp\left(T(x)^\top\theta - F(\theta) + k(x)\right)$ where $\theta \in \Theta$ is the natural parameter space and $A(\theta)$ is the log partition function. Lemma 1 in [56] shows that

$$\chi^2(p_{\theta_1}||p_{\theta_2}) = e^{(A(2\theta_2-\theta_1)-(2A(\theta_2)-A(\theta_1)))} - 1,$$

which is bounded as long as $2\theta_2 - \theta_1 \in \Theta$. This is satisfied for mixture of 1-D Poisson distributions which can be written as $p(w|\lambda) = \frac{1}{w!}\exp(w\log\lambda - \lambda)$ with natural parameter space $\mathbb{R}$, and mixture of $e$-dimensional Multinomial distributions $p(w|\pi) = \exp\left(\langle w, \log\left(\pi/\left(1 - \sum_{i=1}^{e-1}\pi_i\right)\right)\rangle + \log\left(1 - \sum_{i=1}^{e-1}\pi_i\right)\right)$ with natural parameter space $R^{e-1}$. When applied to Gaussian family the natural parameters are

$$\theta_q = \begin{pmatrix} \Sigma_q^{-1}\mu_q \\ vec\left(-\frac{1}{2}\Sigma_q^{-1}\right) \end{pmatrix}.$$

Since the covariance has to be positive definite matrices, $2\theta_q - \theta_{q'}$ may not be a set of valid natural parameter. We deal with this in the next case.

- When $\{P_W^q\}_{q\in\mathcal{Q}}$ is a mixture of isotropic Gaussians, each with mean $\mu_q \in \mathbb{R}^e$ and co-variance $\Sigma_q = \sigma_q^2 I_e$, each satisfying Poincaré inequality with constant $C^q$ (in the single-Gaussian case above we know that $C^q \le \sigma_q^2$), $P_W$ also satisifes Poincaré inequality. We prove this via induction. The key lemma is below:

**Lemma 8** (Corollary 1 of [67]). *Suppose measure $p_0$ is absolutely continuous with respect to measure $p_1$, and $p_0$, $p_1$ satisfy Poincaré inequality with constants $C_0$, $C_1$ respectively, then for all $\alpha \in [0,1]$ and $\beta = 1 - \alpha$, mixture measure $p = \alpha p_0 + \beta p_1$ satisfies Poincaré inequality with with $C \le \max\{C_0, C_1(1 + \alpha\chi_1)\}$ where $\chi_1 = \int \frac{dp_0}{dp_1}dp_0 - 1$.*

We sort the components in the order of non-decreasing $\sigma_q^2$, and add in each component one by one. For each new component $i = 2, \ldots, |\mathcal{Q}|$, we apply the above lemma with $p_0$ being mixture of $P_W^1, \ldots, P_W^{i-1}$ and $p_1$ being the new component $P_W^i$. We only need to prove that $\chi_1$ is bounded at every step. Suppose $p_0 = \sum_{j=1}^{i-1}\alpha_j P_W^j$ with $\sum_{j=1}^{i-1}\alpha_j = 1$, $p_1 = P_W^i$, and $P_W^j = \frac{1}{(2\pi)^{e/2}\sigma_j^e}\exp\left\{-\frac{1}{2}(w-\mu_j)^\top\Sigma_j^{-1}(w-\mu_j)\right\}$. Therefore

$$\chi_1 + 1 = \int \frac{dp_0}{dp_1}dp_0 = \int_w \frac{p_0(w)^2}{p_1(w)}dw$$

$$= \int_w \frac{\sum_{j=1}^{i-1}\frac{\alpha_j^2}{\sigma_j^{2e}}\exp\left\{-\frac{\|w-\mu_j\|^2}{\sigma_j^2}\right\} + \sum_{j=1}^{i-1}\sum_{j'\ne j}\frac{2\alpha_j\alpha_{j'}}{\sigma_j^e\sigma_{j'}^e}\exp\left\{-\frac{\|w-\mu_j\|^2}{2\sigma_j^2} - \frac{\|w-\mu_{j'}\|^2}{2\sigma_{j'}^2}\right\}}{\frac{(2\pi)^{e/2}}{\sigma_i^e}\exp\left\{-\frac{\|w-\mu_i\|^2}{2\sigma_i^2}\right\}}dw$$

The convergence condition of the above integral is $2\sigma_i^2 \ge 2\sigma_j^2$ for all $j < i$ which is satisfied when $\sigma_i^2 \ge \sigma_j^2$.

$\square$

Next we observe that

$$\nabla_w \mathbb{E}[g \mid w] = \nabla_w \int_{x,\tilde{y}}(f(w,x) - \tilde{y})^2 p(x,\tilde{y})dxd\tilde{y} = 2\int_{x,y}(f(w,x) - \tilde{y})\frac{\partial f}{\partial w}p(x,\tilde{y})dxd\tilde{y} = 2\mathbb{E}\left[(f(w,x) - \tilde{y})\frac{\partial f}{\partial w}\right].$$

Since $|f(w,x) - \tilde{y}| \le 1 + \epsilon$ almost surely, $\mathbb{E}\left[\left\|\frac{\partial f}{\partial w}\right\|_2^2\right] \le \beta^2$,

$$\mathbb{E}_h\left[\|\nabla_w\mathbb{E}[g \mid w]\|_2^2\right] = 4\mathbb{E}\left[\left\|(f(w,x) - y)\frac{\partial f}{\partial w}\right\|_2^2\right] \le 4(1+\epsilon)^2\beta^2.$$

Therefore

$$Var_w[\mathbb{E}[g \mid w]] \le C\mathbb{E}\left[\|\nabla_w\mathbb{E}[g \mid w]\|_2^2\right] \le 4(1+\epsilon)^2 C\beta^2.$$

To bound the second term of equation (11), we use concentration of mean of $m$ *i.i.d.* random variables.

Conditioned on $w^{(j)}$, each of the loss $(f(w^{(j)}, x_i^{(j)}) - \tilde{y}_i^{(j)})^2$ are *i.i.d.* and bounded in $[0, (1+\epsilon)^2]$. Hence each variable has variance upper bound $((1+\epsilon)^2 - 0)^2/4 = (1+\epsilon)^4/4$ and the mean has variance upper bound $(1+\epsilon)^4/4m$.

Therefore $Var[g] \leq 4(1+\epsilon)^2 C\beta^2 + (1+\epsilon)^4/4m$. $\qquad\square$

*Proof of Theorem 4.*

$$L(\hat{f}) - L(f^*) \leq 2 \sup_{f \in \mathcal{F}} |\widetilde{L}(f) - \hat{L}(f)|$$

$$\leq 2\mathbb{E} \sup_f |\widetilde{L}(f) - \hat{L}(f)| + 4\sqrt{\frac{\mathbb{E} \sup_f \left|\hat{L}(f) - \widetilde{L}(f)\right| \log(1/\delta)}{n}} + \sqrt{\frac{(32(1+\epsilon)^2 C\beta^2 + \frac{2(1+\epsilon)^4}{m}) \log(1/\delta)}{n}} + \frac{2 \log(1/\delta)}{3n}$$

$$(13)$$

by equation (10) and Lemma 6.

We now show that $\mathbb{E} \sup_f |\widetilde{L}(f) - \hat{L}(f)| \leq 2(1+\epsilon) R_{nm}(F)$. This is similar to the argument for classical Rademacher complexity

$$\mathbb{E}_{w,x,\tilde{y}} \sup_f \left( \frac{1}{nm} \sum_{i,j} (f(w^{(j)}, x_i^{(j)}) - \tilde{y}_i^{(j)})^2 - \mathbb{E}_{w,x,\tilde{y}} (f(w^{(j)}, x_i^{(j)}) - \tilde{y}_i^{(j)})^2 \right)$$

$$\leq \frac{1}{nm} \mathbb{E}_{S,S'} \sup_f \left( \sum_{i,j} [(f(w^{(j)}, x_i^{(j)}) - \tilde{y}_i^{(j)})^2 - (f(w'^{(j)}, x_i'^{(j)}) - \tilde{y}_i'^{(j)})^2] \right)$$

$$= \frac{1}{nm} \mathbb{E}_{S,S',\sigma} \sup_f \left( \sum_{i,j} [\sigma_i^j (f(w^{(j)}, x_i^{(j)}) - \tilde{y}_i^{(j)})^2 - \sigma_i^j (f(w'^{(j)}, x_i'^{(j)}) - \tilde{y}_i'^{(j)})^2] \right)$$

$$\leq \frac{1}{nm} \mathbb{E}_{S,\sigma} \sup_f \left( \sum_{i,j} \sigma_i^j (f(w^{(j)}, x_i^{(j)}) - \tilde{y}_i^{(j)})^2 \right) + \frac{1}{nm} \mathbb{E}_{S',\sigma} \sup_f \left( \sum_{i,j} \sigma_i^j (f(w'^{(j)}, x_i'^{(j)}) - \tilde{y}_i'^{(j)})^2 \right)$$

$$= 2R_{nm}(\widetilde{\mathcal{L}}).$$

where the first inequality uses Jensen's inequality and convexity of $\sup$.

Now we prove the equivalent of Talagrand's contraction lemma to show that $R_{nm}(\widetilde{\mathcal{L}}) \leq 2R_{nm}(\mathcal{F})$. Note that the squared loss is $2(1+\epsilon)$-Lipschitz since $\left| \frac{\partial (f - \tilde{y})^2}{\partial f} \right| = 2|f - \tilde{y}| \leq 2(1+\epsilon)$. We use the following lemma to prove this:

**Lemma 9** (Lemma 5 of [51]). *Suppose* $\{\phi_i\}$, $\{\psi_i\}$, $i = 1, \ldots, N$ *are two sets of functions on* $\Theta$ *such that for each $i$ an $\theta, \theta' \in \Theta$,* $|\phi_i(\theta) - \phi_i(\theta')| \leq |\psi_i(\theta) - \psi_i(\theta')|$. *Then for all functions $c$: $\Theta \to \mathbb{R}$,*

$$\mathbb{E}_\sigma \left[ \sup_\theta \left\{ c(\theta) + \sum_{i=1}^N \sigma_i \phi_i(\theta) \right\} \right] \leq \mathbb{E}_\sigma \left[ \sup_\theta \left\{ c(\theta) + \sum_{i=1}^N \sigma_i \psi_i(\theta) \right\} \right]$$

For any set of $w, x$, we apply Lemma 9 with $\Theta = \mathcal{F}$, $\theta = f$, $N = nm$, $\phi_{ij}(f) = (f(w^{(j)}, x_i^{(j)}) - \tilde{y}_i^{(j)})^2$, $\psi_{ij}(f) = 2(1+\epsilon) f(w^{(j)}, x_i^{(j)})$, and $c(\theta) = 0$. Since $|(f - \tilde{y})^2 - (f' - \tilde{y})^2| \leq 2(1+\epsilon)|f - f'|$, so the condition for Lemma 9 hold. We take expectation over $w, x$ and divide both sides by $nm$ to get

$$\frac{1}{nm} \mathbb{E}_{w,x,\sigma} \sup_f \sum_{j=1}^n \sum_{i=1}^m \sigma_i^j (f(w^{(j)}, x_i^{(j)}) - \tilde{y}_i^{(j)})^2 \leq \frac{2(1+\epsilon)}{nm} \mathbb{E}_{w,x,\sigma} \sup_f \sum_{j=1}^n \sum_{i=1}^m \sigma_i^j f(w^{(j)}, x_i^{(j)})$$

which means $R_{nm}(\mathcal{L}) \leq 2(1+\epsilon) R_{nm}(\mathcal{F})$. Substituting this into inequality (13) finishes the proof.

$\qquad\square$

### A.4 Zero-shot Rademacher complexity bound for the linear hypothesis class

Consider the linear classifier $F = \{(w_1^\top w + w_2^\top x : \|w_1\|_2 \leq B, \|w_1\|_2 \leq C\}$. Suppose $\|w\|_2 \leq 1$ and $\|x\|_2 \leq 1$.

$$
\begin{aligned}
R_{nm}(F) &= \frac{1}{nm}\mathbb{E}_{\sigma,w,x}\sup_w \left\{ \langle w_1, \sum_{ij} \sigma_i^j w^{(j)} \rangle + \langle w_2, \sum_{ij} \sigma_i^j x_i^{(j)} \rangle \right\} \\
&= \frac{1}{nm}\left( B_1 \mathbb{E}_{\sigma,w}\|\sum_{ij}\sigma_i^j w^{(j)}\|_2 + B_2 \mathbb{E}_{\sigma,x}\|\sum_{ij}\sigma_i^j x_i^{(j)}\|_2 \right) \\
&\leq \frac{1}{nm}\left( B_1 \sqrt{m\sum_j \|w^{(j)}\|_2^2} + B_2\sqrt{\sum_{ij}\|x_i^{(j)}\|_2^2} \right) \\
&= (B_1 + B_2)/\sqrt{nm}.
\end{aligned}
$$

We observe that the bound is the same as the standard Rademacher complexity for $nm$ independent samples, which is interesting. The relationship between standard and zero-shot Rademacher complexity for other function classes is an important future direction.

## B  Extended Related Work

Our approach to zero-shot prediction of intervention effects is related to recent advances in heterogenous treatment effect (HTE) estimation, zero-shot learning, and meta-learning.

### B.1  Heterogenous treatment effect (HTE) estimation

**Conditional average treatment effect (CATE) estimation.** A number of approaches have been developed to predict the effect of an existing intervention on an individual or subgroup, based on historical data from individuals who received it. This problem is often referred to in the literature as heterogeneous treatment effect (HTE) estimation [28, 13], to denote that the goal is to detect heterogeneities in how individuals respond to an intervention. A more specific instance of HTE estimation, which we focus on here, is conditional average treatment effect (CATE) estimation [81, 44], in which the goal is to predict the effect of a treatment *conditioned* on an individual's features. A variety of methods and specific models have been developed to achieve this goal [28, 34, 23, 30, 81, 70, 1, 89, 26, 91, 27, 16, 14, 44, 36, 13, 3], and we refer to Bica et al. and Curth et al. for a detailed review of these methods [5, 16]. These methods estimate CATE for an existing intervention, based on historical data from individuals who received it and those that did not.

While these approaches have a number of useful applications, they do not address CATE for novel interventions which did not exist during training (zero-shot). Our primary contribution is a meta-learning framework to leverage these existing CATE estimators for zero-shot predictions. In the CaML framework (Figure 2), each task corresponds to predicting CATE for a single intervention. We synthesize a task by sampling a natural experiment for each intervention, and then use any existing CATE estimator to generate a noisy target label for our the task (Step 2: estimate pseudo-outcomes). We rely on pseuodo-outcome estimates as training labels because prior work has shown that training on observed outcomes directly leads to biased CATE estimates [10, 44, 36], a result which we find holds true in our experiments as well (see T-learner and S-learner w/ meta-learning in Tables 2 and 3).

**Pseudo-outcome estimators.** Prior work has developed a variety of methods to estimate CATE pseudo-outcomes, which are noisy but unbiased estimates of CATE, such as the X-learner [44], R-learner [55], DR-learner [36], and RA-learner [16]. Moreover, the outputs of any other CATE estimation method, such as methods which directly estimate CATE via an end-to-end neural network [34, 70, 72] are an equally valid choice of pseudo-outcome. The literature on pseudo-outcome estimation is growing continuously as new estimators are being developed [21, 40]. Typically, these estimators are specific to a *single binary intervention*, for which a set of nuisance models are trained and used to compute the pseuodo-outcomes. As such, applying meta-learning algorithms to these pseuodo-outcomes requires synthesizing a natural experiment for each intervention, which corresponds to a single task in the CaML  framework.

**Multi-cause estimators.** Our methods to address zero-shot CATE estimation for combinations of interventions are distinct from multi-cause estimators for combinations of binary or categorical interventions [83, 61, 65]. Recent work has shown that these methods can predict the effects of new combinations of interventions [50], when every intervention in the combination has been observed at some point during. However, these methods do not estimate CATE for novel interventions which did not exist during training. By contrast, CaML estimates CATE for zero-shot intervention combinations in which none of the interventions in the combo was ever observed during training (Appendix Table C).

## B.2 Zero-shot learning

Zero-shot learning (ZSL) has traditionally aimed to reason over new concepts and classes [85, 63] which did not exist during training time. While ZSL has primarily focused on natural language processing and computer vision [82], recent interest has been sparked in generalizing over novel interventions (zero-shot) in the biomedical domain [64, 29] in which data can be cheaply collected for hundreds or thousands of possible interventions [92, 75, 19]. However, general-purpose zero-shot causal methods have been largely unexplored. Notable exceptions include GranITE [25] and SIN [25], which each extend a specific CATE estimation [55, 44] method to incorporate intervention information ($W$). However, these approaches have significant drawbacks, which we discuss in Section 2.

## B.3 Meta-learning

Meta-learning, or *learning to learn*, aims to train models which can quickly adapt to new settings and tasks. The key idea is to enable a model to gain experience over multiple learning episodes - in which episodes typically correspond to distinct tasks - to accelerate learning in subsequent learning episodes [32]. The meta-learning literature is rich and spans multiple decades [76, 68, 66, 4], with recent interest focused on model-agnostic methods to train deep learning models to quickly adapt to new tasks [20, 62, 54]. A common focus in the meta-learning literature is few-shot learning, in which a model must adapt to a new task given a small support set of labeled examples. By contrast, we focus on the zero-shot setting, in which no such support set exists. However, we hypothesize that the typical meta-learning problem formulation and training algorithms may also improve zero-shot performance. Thus, CaML's problem formulation and algorithm inspiration from the meta-learning literature, particularly the Reptile algorithm [54] and its application to other tasks in causal inference [71]. Our experimental results show that this meta-learning formulation improves CaML's performance, compared to a standard multi-task learning strategy.

# C   Experimental details

## C.1   Experimental setup

Here, we provide more details about the experimental setup for each investigated setting. This serves to complement the high-level overview given in Table 1. Experiments were run using Google Cloud Services. Deep learning-based methods (i.e., CaML and its ablations, S-learner w/ meta-learning, T-learner w/ meta-learning, SIN, GraphITE, FlexTENET, TARNet, and DragonNet) were run on n1-highmem-64 machines with 4x NVIDIA T4 GPU devices. The remaining baselines (RA-learner, R-learner, X-learner, and T-learner) were run on n1-highmem-64 machines featuring 64 CPUs.

**Fair comparison.** We perform hyper-parameter optimization with random search for all models, with the meta-testing dataset predetermined and held out. To avoid "hyperparameter hacking", hyperparameters ranges are consistent between methods wherever possible, and were chosen using defaults similar to prior work [35, 25]. Choice of final model hyper-parameters was determined using performance metrics (specific to each dataset) computed on the meta-validation dataset, using the best hyper-parameters over 48 runs (6 servers x 4 NVIDIA T4 GPUs per server x 2 runs per GPU ) (Appendix C.4). All table results are computed as the mean across 8 runs of the final model with distinct random seeds.

### C.1.1 Claims dataset

**Interventions** ($W$): We consider drug prescriptions consisting of either one drug, or two drugs prescribed in combination. We observed 745 unique single drugs, and 22,883 unique drug pairs, excluding interventions which occurred less than 500 times. Time of intervention corresponds to the *first* day of exposure. To obtain intervention information, we generated pre-trained drug embeddings from a large-scale biomedical knowledge graph [8] (see Appendix C.5). Drugs correspond to nodes in the knowledge graph, which are linked to other nodes (*e.g.* genes, based on the protein target of the drug). Drug combination embeddings are the sum of the embeddings for their constituent drugs.

**Control group.** A challenge in such causal analyses of clinical settings is defining a control group. We randomly sample 5% (1.52M patients) to use as controls, with a 40/20/40 split betweem meta-train/meta-val/meta-test. When sampling a natural experiment for a given intervention, we select all patients from this control group that did not receive such an intervention. An additional challenge is defining time of intervention for the control group. It is not possible to naively sample a random date, because there are large quiet periods in the claims dataset in which no data is logged. We thus sample a date in which the control patient received *a random drug*, and thus our measure of CATE estimates the *increase* in side effect likelihood from the drug(s) $W$, compared to another drug intervention chosen at random.

**Outcome** ($Y$): We focus on the side effect pancytopenia: a deficiency across all three blood cell lines (red blood cells, white blood cells, and platelets). Pancytopenia is life-threatening, with a 10-20% mortality rate [38, 43], and is a rare side effect of many common medications [42] (*e.g.* arthritis and cancer drugs), which in turn require intensive monitoring of the blood work. Our outcome is defined as the (binary) occurrence of pancytopenia within 90 days of intervention exposure.

**Features** ($X$): Following prior work [24], patient medical history features were constructed by time-binned counts of each unique medical code (diagnosis, procedure, lab result, drug prescription) before the drug was prescribed. In total, 443,940 features were generated from the following time bins: 0-24 hours, 24-48 hours, 2-7 days, 8-30 days, and 31-90 days, 91-365 days, and 365+ days prior. All individuals in the dataset provided by the insurance company had at least 50 unique days of claims data.

**Metrics**: We rely on best practices for evaluating CATE estimators as established established by recent work [86, 11], which recommend to assess treatment rules by comparing subgroups across different quantiles of estimated CATE. We follow the high vs. others RATE (rank-weighted average treatment effect) approach from Yadlowsky et. al [86], which computes the difference in average treatment effect (ATE) of the top $u$ percent of individuals (ranked by predicted CATE), versus all individuals:

$$RATE @ u = \mathbb{E}\Big[Y(1) - Y(0) \mid F_S(S(X)) \geq 1 - u\Big] - \mathbb{E}\Big[Y(1) - Y(0)\Big], \qquad (14)$$

where $S(\cdot)$ is a priority score which ranks samples lowest to highest predicted CATE, and $F_S(\cdot)$ is the cumulative distribution function (CDF) of $S(X_i)$. For instance, RATE @ 0.99 would be the difference between the top 1% of the samples (by estimated CATE) vs. the average treatment effect (ATE) across all samples, which we would expect to be high if the CATE estimator is accurate. The real-world use case of our model would be preventing drug prescription a small subset of high-risk individuals. Thus, more specifically, for each task $j$, intervention $w_j$ in the meta-dataset, and meta-model $\Psi_\theta$ (our priority score $S(\cdot)$), we compute $RATE @ u$ for each $u$ in $[0.999, 0.998, 0.995, 0.99]$ across individuals who received the intervention.

We now summarize how to estimate RATE performance metrics for a single intervention (task). As RATE performance is calculated separately per-intervention we are concerned with a single intervention, we use the simplified notation (i.e. $Y_i(1)$ instead of $Y_i(w)$) from Section 3. Due to the fundamental problem of causal inference (we can only observe $Y_i(0)$ or $Y_i(1)$ for a given sample), the true RATE, as defined above, cannot be directly observed.

We follow the method outlined in Section 2.2 and 2.4 of Yadlowsky et. al, [86] in which we compute $\widehat{\Gamma}_i$, a (noisy but unbiased) estimate for CATE which is in turn used to estimate RATE:

$$\mathbb{E}\Big[\widehat{\Gamma}_i \,\big|\, X_i\Big] \approx \tau(X_i) = \mathbb{E}\Big[Y_i(1) - Y_i(0) \,\big|\, X_i\Big]. \qquad (15)$$

Our data is observational, and as such we can estimate $\widehat{\Gamma}_i$ using a direct non-parametric estimator [80]:

$$\widehat{\Gamma}_i = W_i(Y_i - \hat{m}(X_i, 0)) + (1 - W_i)(\hat{m}(X_i, 1) - Y_i) \tag{16}$$

$$m(x, w) = \mathbb{E}\left[Y_i(w)|X_i = x\right] \tag{17}$$

where $m(x, w)$ is a model that predicts the outcome. Here $\hat{m}(x, w)$ represent nonparametric estimates of $m(x, w)$, respectively, which we obtain by fitting a cross-fitting a model to the intervention natural experiment over 5-folds. We use random forest models for $\hat{m}(x, w)$, as they perform well (achieving $\geq 0.90$ ROC AUC across all meta-testing tasks for predicting outcomes) and are robust to choice of hyperparameters.

RATE can then be estimated via sample-averaging estimator. Specifically, we compute the difference between the average value of $\widehat{\Gamma}_i$ for those in the top $u$ percent of individuals (based on our meta-model's predictions), compared to the average $\widehat{\Gamma}_i$ across all individuals. For further discussion on estimating RATE, we refer readers to [86]. Note that estimates of RATE are *unbounded*: RATE can be less than 0 (due to predictions inversely relating to CATE).

Finally, because our meta-testing dataset consists of individuals treated with drugs *known* in the medical literature to cause pancytopenia (identified by filtering drugs using the side effect database SIDER [42]), observational metrics of recall and precision are also a rough *proxy* for successful CATE estimation. Thus, as secondary metrics, we also compute $Recall @ u$ and $Precision @ u$ for the same set of thresholds as RATE, where a positive label is defined as occurrence of pancytopenia after intervention. We find that these metrics are highly correlated to RATE in our performance results.

**Training & Evaluation:** For each method, we ran a hyperparameter search with 48 random configurations (48 due to running 8 jobs in parallel on 6 servers each) that were drawn uniformly from a pre-defined hyperparameter search space (see Appendix C.4). Methods that can be trained on multiple tasks to then be applied to tasks unseen during training (i.e., CaML and its ablations, S-learner w/ meta-learning, T-learner w/ meta-learning, SIN, GraphITE) were trained for 24 hours (per run) on the meta-training tasks. Model selection was performed on the meta-validation tasks by maximizing the mean RATE@0.998 across meta-validation tasks. Then, the best hyperparameter configuration was used to fit 8 repetition runs across 8 different random seeds. Each repetition model was then tested on the meta-testing tasks, where for all metrics averages across the testing tasks are reported. To make the setting of multi-task models comparable with single-task models that were trained on meta-testing tasks (requiring a train and test split of each meta-testing task), the evaluation of all models was computed on the test split of the meta-testing tasks, respectively. Single-task baselines (FlexTENET, TARNet, and DragonNet, RA-learner, R-learner, X-learner, and T-learner) were given access to the meta-testing tasks during training. Specifically, model selection was performed on the meta-validation tasks, while the best hyperparameter configuration was used to train 8 repetition models (using 8 random seeds) on the train split of each meta-testing task. For the final evaluation, each single-task model that was fit on meta-testing task $i$ was tested on the test split of the same meta-testing task $i$, and the average metrics were reported across meta-testing tasks.

### C.1.2 LINCS

**Interventions ($W$):** Interventions in the LINCS dataset consist of a single perturbagen (small molecule). For intervention information, we used the molecular embeddings for each perturbagen using the RDKit featurizer The same cell line-perturbagen combinations are tested with different perturbagen dosages and times of exposure. [46].To maintain consistency in experimental conditions while also ensuring that the dataset is sufficiently large for training a model, we filter for most frequently occurring dosage and time of exposure in the dataset, which are $10\mu M$ and 24 hours, respectively. We use data from 10,322 different perturbagens.

**Control group.** For each perturbagen (at a given timepoint and dose), we use cell lines which did not receive that intervention as the control group.

**Outcomes ($Y$):** We measure gene expression across the top-50 and top-20 landmark differentially expressed genes (DEGs) in the LINCS dataset. Accurately predicting in gene expression in these DEGs is most crucial to the drug discovery process.

**Features** ($X$): We use 19,221 features from the Cancer Cell Line Encyclopedia (CCLE) [22] to describe each cell-line, based on historical gene expression values in a different lab environment. Our dataset consisted of 99 unique cell lines (after filtering for cell-lines with CCLE features).

**Metrics**: A key advantage of experiments on cells is that at evaluation time we can observe both $Y(0)$ and $Y(1)$ for the same cell line $X$, through multiple experiments on clones of the same cell-line in controlled lab conditions. In the LINCS dataset, $Y(0)$ is also measured for all cells which received an intervention. Thus, we can directly compute the Precision Estimation of Heterogenous Effects (PEHE) on all treated cells in our meta-testing dataset. PEHE is a standard metric for CATE estimation performance [30], analogous to mean squared error (MSE).

$$PEHE = \frac{1}{N} \sum_{i=1}^{N} (\tau_i - \hat{\tau}_i)^2 \tag{18}$$

**Training & Evaluation:** For each method, we ran a hyperparameter search with 48 random configurations (48 due to running 8 jobs in parallel on 6 servers each) that were drawn uniformly from a pre-defined hyperparameter search space (see Appendix C.4). Methods that can be trained on multiple tasks to then be applied to tasks unseen during training (i.e., CaML and its ablations, S-learner w/ meta-learning, T-learner w/ meta-learning, SIN) were trained for 12 hours (per run) on the meta-training tasks. Model selection was performed on the meta-validation tasks by minimizing the overall PEHE for the Top-20 most differentially expressed genes (DEGs) across meta-validation tasks. Then, the best hyperparameter configuration was used to fit 8 repetition runs across 8 different random seeds. Each repetition model was then tested on the meta-testing tasks, where for all metrics averages across the testing tasks are reported.

**Data augmentation:** We augment each batch of data during training to also include treated samples that have their pseudo-outcome labels to 0, and their $W$ set to the zero vector.

## C.2 Selecting holdout interventions for meta-validation and meta-testing

### C.2.1 Claims.

In the 30.4 million patient insurance claims dataset, each intervention task in meta-train/meta-val/meta-testing corresponds to a natural experiment of multiple patients, with some interventions (*e.g.* commonly prescribed drugs) having millions of associated patients who were prescribed the drug. One challenge is that in this setting, there is overlap in subjects between the natural experiments sampled by CaML, which can lead to data leakage between training and testing. For instance, if a patient received Drug 1 (in meta-test) and Drug 2 (meta-train), they would appear in both natural experiments, resulting in data leakage.

We take a conservative approach and exclude all patients who have ever received a meta-testing drug in their lifespan from the natural experiments for meta-val/meta-train. Similarly, we exclude all patients who received a meta-validation drug from meta-training.

This approach means we must take great care in selecting meta-testing drugs. Specifically, we must trade off between selecting drugs that are important (covering enough patients) while not diminishing the training dataset size. For instance selecting a commonly prescribed (*e.g.* aspirin) for meta-testing would deplete our meta-training dataset by over 50% of patients. Thus we only selected meta-test/meta-validation drugs which were prescribed to between 1,000,000 and 100K patients in our dataset, after filtering for only drugs which known to cause Pancytopenia [42] (using the SIDER database). From this subset of drugs, we randomly selected 10 meta-testing drugs and 2 meta-validation drugs, resulting in a total meta-testing/meta-validation pool of 4.1 million patients and 685K patients respectively.

To evaluate on unseen pairs of drugs on the same hold-out test dataset, we additionally created a second pairs testing dataset from the 5 most frequently occurring combinations from the meta-testing dataset. This allowed us to train a single model on the same meta-train split and evaluate on both single drug and drug pair interventions without occurrence of data leakage. Designing a larger evaluation of pairs was not possible because while pairs of drugs are commonly prescribed as intervention, each particular pair of drugs is a rare event, and accurately evaluating CATE estimation performance (for a rare outcome such as Pancytopenia) requires amassing a natural experiment with at least several thousand patients who received the same intervention.

### C.2.2 LINCS.

The goal in selecting holdout interventions for the meta-validation and meta-testing sets was to ensure that they consisted of both cell lines and tasks (small molecules) that had not been seen previously at the time of training (i.e. zero-shot on cell lines and tasks).

Using a random data splitting approach would result in large portions (up to 50%) of the data being unused to comply with the zero-shot requirements on cell lines and tasks. One approach to tackle this was to reserve only those tasks in the held-out sets which had been tested on the fewest cell lines. This preserved the maximum amount of data but resulted in an average of just 1 cell line per task in the meta-testing and meta-validation sets, which would not be fair to the non-zero shot baselines.

To address these issues, we designed a new data split procedure that exploits the structure of how tasks and cell lines are paired. To do so, We first clustered tasks by the cell lines they are tested on. We then identified a set of 600 drugs that had all been tested on a shared set of roughly 20 cell lines. We divided the cell lines and tasks within this set into the meta-validation and meta-testing set, while enforcing zero-shot constraints on both. This resulted in roughly 10 cell lines per intervention in both the meta-validation and meta-testing sets, while still maintaining a reasonably large size of 11 distinct cell lines and 300 distinct tasks in both sets. All remaining tasks and cell lines were reserved for the training set. (See Table 8)

### C.3 Understanding CaML's performance

Our comparison to CATE estimators which are restricted to single interventions (Grey, Table 2,5) shows that a key reason for CaML's strong performance is the ability to joinly learn across from many intervention datasets, in order to generalize to unseen intervention.

Additionally, in both the Claims and LINCS settings, we conduct two key ablation studies to further understand the underlying reason for CaML's strong performance results.

In our first ablation experiment (w/o meta-learning), we trained the CaML model without employing meta-learning, instead using the standard empirical risk minimization (ERM) technique [78]. This can be seen as a specific implementation of the CaML algorithm (refer to Algorithm 1) when $k = 1$ [54]. The results of this experiment showed a varying degree of performance deterioration across our primary tests. In the Claims settings, we observed a decrease in the RATE performance metric by 15%-22% (refer to Table 2), while in the LINCS settings, the PEHE performance metric decreased by approximately 0.01 (see Table 3). These results indicate that the absence of meta-learning affects the model's performance, although the impact varies depending on the specific setting. An important detail to consider is that the Claims data experiments dealt with substantially larger datasets, each comprising hundreds of thousands of patients per intervention. This extensive scale of data potentially amplifies the benefits of using meta-learning in the CaML model for the Claims dataset. The larger dataset enables the model to adapt to a given task over a larger set of iterations without reusing the same data, thereby enhancing the efficacy of meta-learning.

Our second ablation (w/o RA-learner) assesses the sensitivity of CaML's performance to different pseudo-outcome estimation strategies. A key aspect of CaML is *flexibility* in choice of any pseudo-outcome estimator to infer CATE, in contrast to prior work which uses specific CATE estimation strategies [25, 35]. We find that CaML performance benefits strongly from flexibility of pseudo-outcome estimator choice. We assess this by using an alternative pseudo-outcome estimator. Firstly, we find that this ablation results in much noisier model training. For instance, the standard deviation in RATE across the 8 random seeds increases by $20\times$ when using the alternative pseudo-outcome estimator in the claims setting. Moreover, the alternative pseudo-outcome estimator typically worsens performance, decreasing RATE by up to 6% in the Claims setting , and increasing PEHE by 20%-21% in the LINCS setting (Table 3). We note that this ablation performs slightly better at the 0.99 threshold, which may be a result of the high variance in this ablation. Specific choice of alternative pseudo-outcome estimator for this ablation varies by setting. We use the R-learner [55] for Claims as it also achieves strong single task performance (Table 2, grey) on Claims data. However, R-learner is restricted to single-dimensional outcomes, and thus for LINCS (in which outcomes are 50 and 20 dimensional), we use the PW-learner instead [16].

## C.4 Hyperparameter space

### C.4.1 Claims dataset hyperparameter space

We list the hyperparameter search spaces for the medical claims dataset in the following tables. Table 9 represents the search space for CaML. The SIN baseline consists of two stages, Stage 1 and Stage 2. For the Stage 1 model, we searched the identical hyperparameter search space as for CaML (Table 9). For Stage 2, we used the hyperparameters displayed in Table 10. The search space for the GraphITE baseline is displayed in Table 11. For the S-learner and T-learner w/ meta-learning baselines, we use the same hyperparameter space as for CaML (Table 9) with the only major difference that the these baselines predicts the outcome $Y$ instead of $\hat{\tau}$. For all deep learning-based methods, we employed a batch size of $8,192$, except for GraphITE, where we were restricted to using a batch size of $512$ due to larger memory requirements. Single-task neural network baselines (FlexTENet, TARNet, and DragonNet) are shown in Tables 12,13, and 14, respectively. For the remaining baselines, i.e., the model-agnostic CATE estimators, the (shared) hyperparameter search space is shown in Table 15. Finally, applied L1 regularization to the encoder layer of the customizable neural network models (that were not reused as external packages), i.e., SIN learner, GraphITE, T-learner w/ meta-learning, and S-learner w/ meta-learning, and CaML.

### C.4.2 LINCS hyperparameter space

We list the hyperparameter search spaces for LINCS in the following tables. CaMLis shown in Table 16. SIN Stage 1 used the same search space as CaML (Table 16. The search space of SIN Stage 2 is shown in Table 17. S learner and T-learner w/ meta-learning used the same search space as CaML. The search space of GraphITE is shown in Table 18. All methods that were applied to LINCS used a batch size of 20.

### C.5 More details on intervention information

Here we give more details about the intervention information used for the medical claims dataset. In order to perform zero-shot generalization, we acquired information about a specific intervention through the use of pretrained embeddings. We generated these embeddings on the Precision Medicine Knowledge Graph [8] that contains drug nodes as well as 9 other node types. We extracted embeddings for 7957 drugs from the knowledge graph.

To extract rich neighborhood information from the knowledge graph we used Stargraph [49], which is a coarse-to-fine representation learning algorithm. StarGraph generates a subgraph for each node by sampling from its neighbor nodes (all nodes in the one-hop neighborhood) and anchor nodes (a preselected subset of nodes appearing in the multihop neighborhood). In our case the anchor nodes were the 2% of graph nodes with the highest degree. For the scoring function we used the augmented version of TripleRE [90] presented in the StarGraph article [49].

We performed a hyperparameter optimization to compare different models and determine the one we used to calculate our final embeddings (see Table C.5). The hyperparameter search was random with the objective of minimizing the loss function used in training on held out data. The search range for each of the parameters is displayed in C.5. Since certain parameters did not seem to influence the final score as much we decided to use them as constants and focus on optimizing the hyperparameters in the table. Therefore the number of sampled anchors was set to 20 and $u = 0.1$ in the augmented TripleRE function, the values matching those seen in Stargraph [48].

Our final embeddings were 256-dimensional, the learning rate was 2e-4, the drop-ratio was 5e-3. We used the self-adversarial negative sampling loss with $\gamma = 8$ and we sampled 4 neighbor nodes for each subgraph.

To additionally evaluate the quality of the embeddings we assigned classes to drug combinations and then scored them using multiple clustering metrics. We were interested to see if embeddings of drug combinations used for similar purposes would be embedded closer together than other drug combinations. For the class label of single drugs we used the first level of the Anatomical Therapeutic Chemical (ATC) code, which represents one of the 14 anatomical or pharmacological groups. Since certain medications have more than one ATC code, we took the mode of all labels for a specific drug. For multiple drugs we combined all distinct first level values and took the mode of them as the label. We used the Silhouette metric, Calinski Harabasz index and Davies Bouldin index as

well as the average classification accuracy over 10 runs of training a random forest classifier on a random sample of 80% of the dataset and evaluating on the remaining 20%. Out of all tested embeddings the hyperparameter optimized StarGraph embeddings performed best (exceeding 93% in the classification accuracy metric).

## C.6 Pseudo-outcome estimation

In our experiments, we estimate pseudo-outcomes $\tilde{\tau}$ for a given intervention $w$ using the RA-learner [16]:

$$\tilde{\tau} = W(Y - \hat{\mu}_0(X)) + (1 - W)(\hat{\mu}_1(X) - Y) \tag{19}$$

where $\hat{\mu}_w$ is an estimate of $\mu_w(X) = \mathbb{E}_{\mathcal{P}}\Big[Y \mid X = x, W = w\Big]$.

Furthermore, in both settings we only estimate CATE for treated individuals. We focus on treated individuals in the Claims setting because we care about the risk of an adverse event for prescribing a sick patients drugs that may cure their sickness, not the adverse event risk of prescribing healthy patients drugs (which is of less clinical interest). In the LINCS setting, we focus on treated cells as for these cell-lines $Y(0)$ is also measured from a cloned cell-line under similar laboratory conditions, which allows us to directly estimate CATE prediction performance using the PEHE metric. As we focus on treated samples, the RA-learner can be simplified to $\tilde{\tau} = Y - \hat{\mu}_0(X)$. We estimate $\hat{\mu}_0(X)$ using a random forest model in the Claims setting, whereas in the LINCS setting we use the point estimate from the untreated control cell line's gene expression.

## C.7 Baselines

Here we provide more details on the baselines used in our experiments.

*Trained on test task:* These baselines leverage CATE estimators which can only be trained on a single task (typically these are the strongest baselines, when there is a large enough dataset for a single task). Thus, we train a single model for each meta-testing task on its train split, and evaluate performance on its test split. We use a number of strong baselines for CATE estimation developed by prior work including both model-agnostic and end-to-end deep learning approaches: T-learner. Specifically, we use the model-agnostic CATE estimators: [44], X-learner [44], RA-learner [16], R-learner [55]. We additionally use the end-to-end deep learning estimators DragonNet [72], TARNet [70], and FlexTENet [17], using implementations from [17]. For model-agnostic CATE estimators, we use random forest models following prior work [14, 81].

*Zero-shot.* These baselines use CATE estimators which incorporate intervention information ($W$) and are capable of multi-task learning. We train these baselines on all meta-training tasks. These baselines have no access to the meta-testing tasks during training. We found in preliminary experiments that in some cases, baseline models trained with vanilla ERM would not even converge. To allow for fair comparison to baselines, we allow for all zero-shot baselines to be trained using Reptile (by training using the same optimization strategy as Algorithm 1, while allowing for training with ERM by including $k = 1$ in the hyperparameter search space).

Firstly, we use GraphITE [25] and Structured Intervention Networks [35]. These are, to the best of our knowledge, the only methods from prior work which are (in principle) capable of zero-shot generalization. We use existing implementations provided by the authors [35].

Additionally, we implement two strong baselines which estimate CATE by modeling $Y(w)$ and $Y(0)$, rather than via pseudo-outcomes. These are variants of the S-learner and T-learner [44] with meta-learning, which use the intervention information as input, rather than one-hot encoded vectors of the different interventions—such that they also have zero-shot capability. Specifically, we train MLPs using the same architecture as CaML to estimate the response function from observed outcomes:

$$\mu(x, w) = \mathbb{E}_{\mathcal{P}}\Big[Y \mid X = x, W = w\Big] \tag{20}$$

and estimate CATE by

$$\hat{\tau}_w(x) = \hat{\mu}(x, w) - \hat{\mu}(x, \mathbf{0}) \tag{21}$$

Where $w$ denotes the corresponding intervention information $w$ for an intervention, and $\mathbf{0}$ denotes a null intervention vector. In the LINCS setting, we represent $\mathbf{0}$ as a vector of zeros, whereas in the

Claims setting we represent **0** as the mean embedding of all drugs (as the estimand is the increase in adverse event likelihood compared to a randomly chosen drug). The difference between the T-learner and the S-learner is that the T-learner estimates two models, one for control units and one for treated units. By contrast, the S-learner estimates a shared model across all units.

# D    Additional Experiments

In general, limited training examples, or biases in the training data, will degrade model performance—and the CaML algorithm is no exception in this regard. For instance, if there are too few examples of prior interventions (e.g., only a handful of training drugs), then zero-shot generalization may become more challenging. Therefore, it is important to study the robustness of CaML's performance to limitations in the training dataset. To this end, we conduct additional experiments in which we downsample the number of training interventions. As expected, we find that: (1) zero-shot capabilities improve as the set of unique training interventions increases in size and (2) we can still achieve strong performance on smaller datasets (e.g., runs with 60% and 80%, of the interventions achieve similar performance).

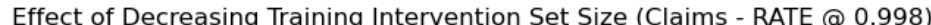

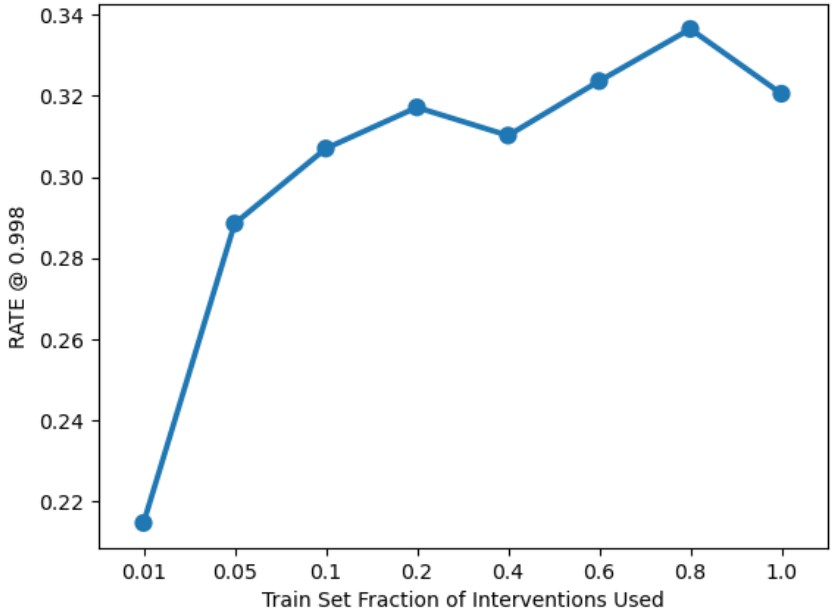

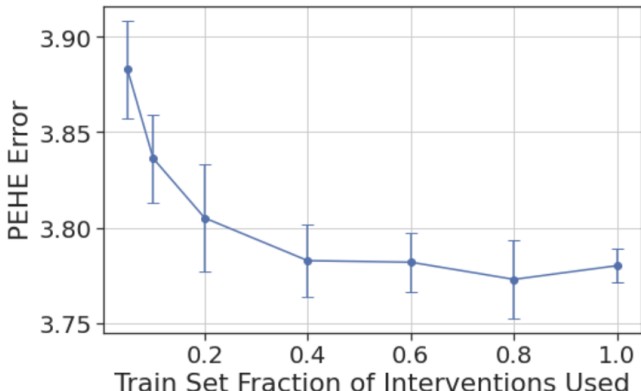

Figure 3: Measuring the robustness of CaML to limitations in the training intervention data. We downsample the number of training interventions and measure CaML's performance. Overall, we find that CaML's zero-shot capabilities improve as the set of unique training interventions increases in size. Nevertheless, CaML still achieves strong performance on smaller datasets (e.g., runs with 60% and 80%, of the interventions achieve similar performance). Results are analogous for other metrics on both datasets. Top: Performance on the Claims dataset at predicting the effect on a novel drug on the likelihood of Pancytopenia onset (RATE @ 0.998). Bottom: Performance on the LINCS dataset at predicting the gene expression of the Top 20 and Top 50 most differentially expressed genes (DEGs).

# E  Unbiasedness of CATE estimates

## Unbiasedness of Pseudo-outcome labels

We show for an example pseudo-outcome label, the RA-learner [16], that the estimated pseudo-outcome labels are unbiased estimates of the true CATE, i.e.:

$$\mathbb{E}\left[\tilde{\tau}|X=x\right] = \tau(x) = \mathbb{E}\left[Y(1) - Y(0)|X=x\right]$$

The pseudo-outcome $\tilde{\tau}$ for the RA-learner is defined as $\tilde{\tau}_i = Y_i - \hat{\mu}_0(X_i)$ for treated units ($W_i = 1$), and $\tilde{\tau}_i = \hat{\mu}_1(X_i) - Y_i$ for control units ($W_i = 0$).

Here, $\hat{\mu}_0(X), \hat{\mu}_1(X)$ denote unbiased and correctly specified nuisance models for the outcomes $Y(0)$ and $Y(1)$ respectively. In other words, $\mathbb{E}\left[\hat{\mu}_0(x)\right] = \mu_0(x) = \mathbb{E}\left[Y(0)|X=x\right]$ and $\mathbb{E}\left[\hat{\mu}_1(x)\right] = \mu_1(x) = \mathbb{E}\left[Y(1)|X=x\right]$.

We consider the treated and control units separately. For treated units ($W_i = 1$), we have:

$$\tilde{\tau}_i = Y_i - \hat{\mu}_0(X_i).$$

Hence, their expectation, conditioned on covariates $X$ can be written as:

$$\mathbb{E}\left[\tilde{\tau}|X=x\right] = \mathbb{E}\left[Y - \hat{\mu}_0(X)|X=x\right] = \mathbb{E}\left[Y|X=x\right] - \mathbb{E}\left[\hat{\mu}_0(X)|X=x\right] = \mathbb{E}\left[Y|X=x\right] - \mathbb{E}\left[Y(0)|X=x\right] = \tau(x),$$

which by applying the consistency assumption (paper Line 98) for treated units is equivelant to:

$$\mathbb{E}\left[Y(1)|X=x\right] - \mathbb{E}\left[Y(0)|X=x\right] = \mathbb{E}\left[Y(1) - Y(0)|X=x\right] = \tau(x).$$

Analogously, we can make the same argument for control units ($W = 0$). Here, the pseudo-outcome is computed as:

$$\tilde{\tau}_i = \hat{\mu}_1(X_i) - Y_i.$$

Hence, we have

$$\mathbb{E}\left[\tilde{\tau}|X=x\right] = \mathbb{E}\left[\hat{\mu}_1(X) - Y|X=x\right] = \mathbb{E}\left[\hat{\mu}_1(X)|X=x\right] - \mathbb{E}\left[Y|X=x\right],$$

which under consistency (for control units) is equivalent to:

$$\mathbb{E}\left[Y(1)|X=x\right] - \mathbb{E}\left[Y(0)|X=x\right] = \mathbb{E}\left[Y(1) - Y(0)|X=x\right] = \tau(x).$$

## Unbiasedness of Model Loss

We consider parametrized CATE estimators $\Psi_\theta \colon \mathbb{R}^e \times \mathbb{R}^d \to \mathbb{R}$ that take as input intervention information $w \in \mathbb{R}^e$ (e.g., a drug's attributes) and individual features $x \in \mathbb{R}^d$ (e.g., patient medical history) to return a scalar for the estimated CATE (e.g., the effect of the drug on patient health).

We denote the loss function $L$ with regard to a CATE estimator $\Psi$ and a target $y$ as:

$$L(\Psi, y) = \left(\Psi\left(w, x\right) - y\right)^2$$

As in Theorem 1, we assume pseudo-outcomes targets $\tilde{\tau}$ during training satisfy $\tilde{\tau} = \tau + \xi$ where $\tau$ is the true (unobserved) CATE and $\xi$ is an independent zero-mean noise.

**Lemma 10.** *Given two different CATE estimators $\hat{\Psi}_{\theta_1}, \hat{\Psi}_{\theta_2}$, parameterized by $\theta_1$ and $\theta_2$:*

$$\mathbb{E}\left[L(\hat{\Psi}_{\theta_1}, \tilde{\tau})\right] \leq \mathbb{E}\left[L(\hat{\Psi}_{\theta_2}, \tilde{\tau})\right] \implies \mathbb{E}\left[L(\hat{\Psi}_{\theta_1}, \tau)\right] \leq \mathbb{E}\left[L(\hat{\Psi}_{\theta_2}, \tau)\right]$$

*Proof.* We follow a similar argument as Tripuraneni et al. [77].

$$\begin{aligned}
\mathbb{E}\left[L(\hat{\Psi}_\theta, \tilde{\tau})\right] &= \mathbb{E}\left[\left(\hat{\Psi}_\theta\left(w, x\right) - \tilde{\tau}\right)^2\right] = \mathbb{E}\left[\left(\hat{\Psi}_\theta\left(w, x\right) - \tau + \tau - \tilde{\tau}\right)^2\right] \\
&= \mathbb{E}\left[\left(\hat{\Psi}_\theta\left(w, x\right) - \tau\right)^2 + (\tau - \tilde{\tau})^2 + 2\left(\hat{\Psi}_\theta\left(w, x\right) - \tau\right)(\tau - \tilde{\tau})\right] \\
&= \mathbb{E}\left[\left(\hat{\Psi}_\theta\left(w, x\right) - \tau\right)^2\right] + \mathbb{E}\left[(\tau - \tilde{\tau})^2\right] + 2\mathbb{E}\left[\left(\hat{\Psi}_\theta\left(w, x\right) - \tau\right)(\tau - \tilde{\tau})\right].
\end{aligned}$$

Now we subtract the loss for the two models parameterized by by $\theta_1$ and $\theta_2$:

$$\mathbb{E}\left[L(\hat{\Psi}_{\theta_1}, \tilde{\tau})\right] - \mathbb{E}\left[L(\hat{\Psi}_{\theta_2}, \tilde{\tau})\right] = \qquad (1)$$

$$\mathbb{E}\left[\left(\hat{\Psi}_{\theta_1}(w,x) - \tau\right)^2\right] + \mathbb{E}\left[(\tau - \tilde{\tau})^2\right] + 2\mathbb{E}\left[\left(\hat{\Psi}_{\theta_1}(w,x) - \tau\right)(\tau - \tilde{\tau})\right] -$$

$$\left(\mathbb{E}\left[\left(\hat{\Psi}_{\theta_2}(w,x) - \tau\right)^2\right] + \mathbb{E}\left[(\tau - \tilde{\tau})^2\right] + 2\mathbb{E}\left[\left(\hat{\Psi}_{\theta_2}(w,x) - \tau\right)(\tau - \tilde{\tau})\right]\right) =$$

$$\mathbb{E}\left[\left(\hat{\Psi}_{\theta_1}(w,x) - \tau\right)^2\right] + 2\mathbb{E}\left[\left(\hat{\Psi}_{\theta_1}(w,x) - \tau\right)(\tau - \tilde{\tau})\right] -$$

$$\mathbb{E}\left[\left(\hat{\Psi}_{\theta_2}(w,x) - \tau\right)^2\right] - 2\mathbb{E}\left[\left(\hat{\Psi}_{\theta_2}(w,x) - \tau\right)(\tau - \tilde{\tau})\right] =$$

Expanding out the righthand terms give us:

$$\mathbb{E}\left[\left(\hat{\Psi}_{\theta_1}(w,x) - \tau\right)^2\right] + 2\mathbb{E}\left[\hat{\Psi}_{\theta_1}(w,x) \cdot \tau - \hat{\Psi}_{\theta_1}(w,x) \cdot \tilde{\tau} - \tau^2 + \tau \cdot \tilde{\tau}\right] -$$

$$\mathbb{E}\left[\left(\hat{\Psi}_{\theta_2}(w,x) - \tau\right)^2\right] - 2\mathbb{E}\left[\hat{\Psi}_{\theta_2}(w,x) \cdot \tau - \hat{\Psi}_{\theta_2}(w,x) \cdot \tilde{\tau} - \tau^2 + \tau \cdot \tilde{\tau}\right] =$$

$$\mathbb{E}\left[\left(\hat{\Psi}_{\theta_1}(w,x) - \tau\right)^2\right] + 2\mathbb{E}\left[\hat{\Psi}_{\theta_1}(w,x) \cdot \tau - \hat{\Psi}_{\theta_1}(w,x) \cdot \tilde{\tau}\right] - \mathbb{E}\left[\tau^2\right] + \mathbb{E}\left[\tau \cdot \tilde{\tau}\right] -$$

$$\mathbb{E}\left[\left(\hat{\Psi}_{\theta_2}(w,x) - \tau\right)^2\right] - 2\mathbb{E}\left[\hat{\Psi}_{\theta_2}(w,x) \cdot \tau - \hat{\Psi}_{\theta_2}(w,x) \cdot \tilde{\tau}\right] + \mathbb{E}\left[\tau^2\right] - \mathbb{E}\left[\tau \cdot \tilde{\tau}\right] =$$

$$\mathbb{E}\left[\left(\hat{\Psi}_{\theta_1}(w,x) - \tau\right)^2\right] + 2\mathbb{E}\left[\hat{\Psi}_{\theta_1}(w,x) \cdot \tau - \hat{\Psi}_{\theta_1}(w,x) \cdot \tilde{\tau}\right] -$$

$$\mathbb{E}\left[\left(\hat{\Psi}_{\theta_2}(w,x) - \tau\right)^2\right] - 2\mathbb{E}\left[\hat{\Psi}_{\theta_2}(w,x) \cdot \tau - \hat{\Psi}_{\theta_2}(w,x) \cdot \tilde{\tau}\right] =$$

$$\mathbb{E}\left[\left(\hat{\Psi}_{\theta_1}(w,x) - \tau\right)^2\right] - \mathbb{E}\left[\left(\hat{\Psi}_{\theta_2}(w,x) - \tau\right)^2\right] + 2\mathbb{E}\left[\left(\hat{\Psi}_{\theta_1}(w,x) \cdot (\tau - \tilde{\tau})\right) - \left(\hat{\Psi}_{\theta_2}(w,x) \cdot (\tau - \tilde{\tau})\right)\right] =$$

$$\mathbb{E}\left[\left(\hat{\Psi}_{\theta_1}(w,x) - \tau\right)^2\right] - \mathbb{E}\left[\left(\hat{\Psi}_{\theta_2}(w,x) - \tau\right)^2\right] + 2\mathbb{E}\left[\left(\hat{\Psi}_{\theta_1}(w,x) - \hat{\Psi}_{\theta_2}(w,x)\right) \cdot \underbrace{(\tau - \tilde{\tau})}_{-\xi}\right] =$$

$$\mathbb{E}\left[\left(\hat{\Psi}_{\theta_1}(w,x) - \tau\right)^2\right] - \mathbb{E}\left[\left(\hat{\Psi}_{\theta_2}(w,x) - \tau\right)^2\right] + 2\mathbb{E}\left[\left(\hat{\Psi}_{\theta_1}(w,x) - \hat{\Psi}_{\theta_2}(w,x)\right) \cdot -\xi\right] =$$

$$\mathbb{E}\left[\left(\hat{\Psi}_{\theta_1}(w,x) - \tau\right)^2\right] - \mathbb{E}\left[\left(\hat{\Psi}_{\theta_2}(w,x) - \tau\right)^2\right] + 2\mathbb{E}\left[\hat{\Psi}_{\theta_1}(w,x) - \hat{\Psi}_{\theta_2}(w,x)\right]\mathbb{E}\left[-\xi\right] =$$

$$\mathbb{E}\left[\left(\hat{\Psi}_{\theta_1}(w,x) - \tau\right)^2\right] - \mathbb{E}\left[\left(\hat{\Psi}_{\theta_2}(w,x) - \tau\right)^2\right] + 2\mathbb{E}\left[\hat{\Psi}_{\theta_1}(w,x) - \hat{\Psi}_{\theta_2}(w,x)\right]\underbrace{\mathbb{E}\left[-\xi\right]}_{0} =$$

$$\mathbb{E}\left[L(\hat{\Psi}_{\theta_1}, \tau)\right] - \mathbb{E}\left[L(\hat{\Psi}_{\theta_2}, \tau)\right],$$

from which—due to equality with Equation 1—the claim follows.

$\square$

| | RATE @$u$ (↑) | | | | Recall @$u$ (↑) | | | | Precision @$u$ (↑) | | | |
|---|---|---|---|---|---|---|---|---|---|---|---|---|
| | 0.999 | .998 | 0.995 | 0.99 | 0.999 | 0.998 | 0.995 | 0.99 | 0.999 | 0.998 | 0.995 | 0.99 |
| Random | 0.00∓<0.001 | 0.00∓<0.001 | 0.00∓<0.001 | 0.00∓<0.001 | 0.00∓<0.001 | 0.00∓<0.001 | 0.01∓<0.001 | 0.00∓<0.001 | 0.00∓<0.001 | 0.00∓<0.001 | 0.00∓<0.001 | 0.00±<0.001 |
| T-learner | 0.32∓<0.001 | 0.26∓<0.001 | 0.16∓<0.001 | 0.10∓<0.001 | 0.12∓<0.001 | 0.18∓<0.001 | 0.26∓<0.001 | 0.31∓<0.001 | 0.36∓<0.001 | 0.29∓<0.001 | 0.18∓<0.001 | 0.11∓<0.001 |
| X-learner | 0.06∓<0.001 | 0.05∓<0.001 | 0.04∓<0.001 | 0.03∓<0.001 | 0.02∓<0.001 | 0.04∓<0.001 | 0.08∓<0.001 | 0.12∓<0.001 | 0.09∓<0.001 | 0.07∓<0.001 | 0.06∓<0.001 | 0.05∓<0.001 |
| R-learner | 0.19∓<0.001 | 0.17∓<0.001 | 0.12∓<0.001 | 0.08∓<0.001 | 0.06∓<0.001 | 0.10∓<0.001 | 0.19∓<0.001 | 0.26∓<0.001 | 0.24∓<0.001 | 0.21∓<0.001 | 0.15∓<0.001 | 0.11∓<0.001 |
| RA-learner | 0.47∓0.001 | 0.37∓<0.001 | 0.23∓<0.001 | 0.14∓<0.001 | 0.17∓<0.001 | 0.26∓<0.001 | 0.38∓<0.001 | 0.45∓<0.001 | 0.54∓0.001 | 0.42∓<0.001 | 0.26∓<0.001 | 0.16∓<0.001 |
| DragonNet | 0.09±0.037 | 0.07±0.030 | 0.05±0.019 | 0.04±0.013 | 0.02±0.008 | 0.04±0.012 | 0.07±0.020 | 0.10±0.027 | 0.12±0.045 | 0.10±0.036 | 0.07±0.023 | 0.05±0.015 |
| TARNet | 0.15±0.011 | 0.12±0.011 | 0.07±0.006 | 0.05±0.004 | 0.05±0.003 | 0.08±0.006 | 0.12±0.008 | 0.14±0.011 | 0.18±0.013 | 0.15±0.012 | 0.09±0.007 | 0.06±0.004 |
| FlexTENet | 0.10±0.015 | 0.09±0.016 | 0.06±0.008 | 0.04±0.006 | 0.04±0.006 | 0.07±0.009 | 0.12±0.011 | 0.17±0.017 | 0.12±0.018 | 0.11±0.019 | 0.08±0.010 | 0.06±0.007 |
| GraphITE | 0.19±0.024 | 0.12±0.013 | 0.05±0.004 | 0.03±0.002 | 0.07±0.009 | 0.08±0.010 | 0.09±0.008 | 0.10±0.008 | 0.23±0.027 | 0.14±0.015 | 0.07±0.005 | 0.04±0.003 |
| SIN | 0.00±0.002 | 0.00±0.001 | 0.00±0.001 | 0.00±0.001 | 0.00±0.001 | 0.00±0.001 | 0.01±0.001 | 0.02±0.002 | 0.01±0.002 | 0.01±0.001 | 0.01±0.001 | 0.01±0.001 |
| S-learner w/ meta-learning | 0.21±0.032 | 0.16±0.028 | 0.09±0.020 | 0.05±0.012 | 0.08±0.013 | 0.11±0.022 | 0.15±0.035 | 0.16±0.038 | 0.25±0.034 | 0.18±0.031 | 0.10±0.023 | 0.06±0.014 |
| T-learner w/ meta-learning | 0.40±0.012 | 0.31±0.010 | 0.18±0.007 | 0.11±0.004 | 0.15±0.006 | 0.22±0.008 | 0.32±0.013 | 0.38±0.014 | 0.45±0.013 | 0.35±0.011 | 0.21±0.008 | 0.13±0.004 |
| CaML - w/o meta-learning | 0.39±0.012 | 0.31±0.006 | 0.18±0.008 | 0.11±0.006 | 0.15±0.005 | 0.22±0.007 | 0.32±0.014 | 0.39±0.021 | 0.45±0.010 | 0.35±0.006 | 0.22±0.008 | 0.14±0.006 |
| CaML - w/o RA-learner | 0.45±0.058 | 0.36±0.066 | 0.22±0.067 | **0.14**±0.041 | 0.16±0.020 | 0.24±0.019 | 0.35±0.016 | 0.41±0.023 | 0.51±0.076 | 0.41±0.082 | 0.26±0.078 | **0.16**±0.048 |
| CaML (ours) | **0.48**±0.010 | **0.38**±0.007 | **0.23**±0.003 | 0.13±0.002 | **0.18**±0.004 | **0.27**±0.005 | **0.38**±0.006 | **0.45**±0.010 | **0.54**±0.012 | **0.43**±0.008 | **0.26**±0.004 | 0.16±0.003 |

Table 4: Performance results for the Claims dataset (predicting pancytopenia onset from drug exposure using patient medical history. This table extends Table 2 with standard deviations.

| | RATE @$u$ (↑) | | | | Recall @$u$ (↑) | | | | Precision @$u$ (↑) | | | |
|---|---|---|---|---|---|---|---|---|---|---|---|---|
| | 0.999 | .998 | 0.995 | 0.99 | 0.999 | 0.998 | 0.995 | 0.99 | 0.999 | 0.998 | 0.995 | 0.99 |
| Random | 0.00±<0.001 | 0.00±<0.001 | 0.00±<0.001 | 0.00±<0.001 | 0.0±<0.001 | 0.0±<0.001 | 0.01±<0.001 | 0.01±<0.001 | 0.01±<0.0014 | 0.01±<0.001 | 0.01±<0.001 | 0.00±<0.001 |
| T-learner | 0.10±<0.001 | 0.07±<0.001 | 0.05±<0.001 | 0.04±<0.001 | 0.05±<0.001 | 0.07±<0.001 | 0.11±<0.001 | 0.13±<0.001 | 0.10±<0.001 | 0.08±<0.001 | 0.06±<0.001 | 0.04±<0.001 |
| X-learner | 0.00±<0.001 | -0.01±<0.001 | 0.00±<0.001 | 0.00±<0.001 | 0.00±<0.001 | 0.00±<0.001 | 0.01±<0.001 | 0.02±<0.001 | 0.00±<0.001 | 0.00±<0.001 | 0.00±<0.001 | 0.01±<0.001 |
| R-learner | -0.01±<0.001 | -0.01±<0.001 | -0.01±<0.001 | 0.00±<0.001 | 0.00±<0.001 | 0.00±<0.001 | 0.00±<0.001 | 0.04±<0.001 | 0.00±<0.001 | 0.00±<0.001 | 0.00±<0.001 | 0.01±<0.001 |
| RA-learner | 0.28±<0.001 | 0.26±<0.001 | 0.17±<0.001 | 0.10±<0.001 | 0.10±<0.001 | 0.19±<0.001 | 0.30±<0.001 | 0.37±<0.001 | 0.30±<0.001 | 0.28±<0.001 | 0.18±<0.001 | 0.11±<0.001 |
| DragonNet | -0.01±0.002 | 0.00±0.009 | 0.00±0.004 | 0.00±0.003 | 0.00±<0.001 | 0.00±0.003 | 0.00±0.005 | 0.01±0.009 | 0.00±<0.001 | 0.00±0.010 | 0.00±0.004 | 0.00±0.003 |
| TARNet | 0.04±0.046 | 0.03±0.030 | 0.02±0.013 | 0.02±0.012 | 0.01±0.011 | 0.02±0.015 | 0.04±0.013 | 0.06±0.029 | 0.05±0.046 | 0.04±0.032 | 0.03±0.013 | 0.02±0.012 |
| FlexTENet | 0.02±0.024 | 0.02±0.019 | 0.04±0.012 | 0.03±0.013 | 0.01±0.009 | 0.03±0.018 | 0.08±0.012 | 0.12±0.037 | 0.02±0.027 | 0.03±0.020 | 0.04±0.012 | 0.04±0.014 |
| S-learner w/ meta-learning | 0.27±0.173 | 0.16±0.118 | 0.08±0.052 | 0.04±0.030 | 0.09±0.055 | 0.10±0.070 | 0.13±0.084 | 0.15±0.090 | 0.29±0.180 | 0.18±0.123 | 0.09±0.055 | 0.05±0.032 |
| T-learner w/ meta-learning | 0.27±0.173 | 0.16±0.118 | 0.08±0.052 | 0.04±0.030 | 0.09±0.055 | 0.10±0.070 | 0.13±0.084 | 0.15±0.090 | 0.29±0.180 | 0.18±0.123 | 0.09±0.055 | 0.05±0.032 |
| GraphITE | 0.25±0.088 | 0.15±0.054 | 0.06±0.025 | 0.03±0.011 | 0.08±0.024 | 0.10±0.034 | 0.11±0.045 | 0.13±0.049 | 0.27±0.091 | 0.16±0.057 | 0.07±0.027 | 0.04±0.013 |
| SIN | 0.00±0.008 | 0.00±0.014 | 0.00±0.008 | 0.00±0.005 | 0.00±0.005 | 0.00±0.008 | 0.02±0.015 | 0.03±0.009 | 0.00±0.007 | 0.01±0.014 | 0.01±0.009 | 0.01±0.005 |
| CaML - w/o meta-learning | 0.45±0.070 | **0.38**±0.057 | 0.21±0.017 | 0.13±0.008 | 0.19±0.019 | 0.28±0.026 | 0.38±0.025 | 0.45±0.019 | 0.49±0.070 | **0.41**±0.057 | 0.23±0.017 | 0.15±0.008 |
| CaML - w/o RA-learner | 0.40±0.101 | 0.33±0.034 | **0.24**±0.014 | 0.15±0.010 | 0.18±0.025 | 0.28±0.010 | 0.42±0.024 | 0.50±0.028 | 0.44±0.099 | 0.36±0.033 | **0.26**±0.014 | 0.17±0.010 |
| CaML (ours) | **0.47**±0.084 | 0.37±0.044 | 0.23±0.022 | **0.15**±0.013 | **0.20**±0.015 | **0.30**±0.016 | **0.43**±0.024 | **0.51**±0.027 | **0.51**±0.079 | 0.40±0.044 | 0.25±0.023 | **0.17**±0.013 |

Table 5: Performance results for the medical claims dataset, in which the task is to predict the effect of a *pair* of drugs the drug on pancytopenia occurrence. Mean and standard deviation between runs is reported. Single-task methods were trained on the meta-testing tasks (best model underlined). Methods that were capable of training across multiple tasks were trained on meta-training tasks and applied to previously unseen meta-testing tasks (best model in bold).CaML outperforms the strongest baseline that had access to testing tasks on 12 out of 12 metrics, and outperforms all zero-shot baselines. Notably, due to the small sample size for natural experiments with combinations of drugs, *the RATE estimation process is very noisy* which is reflected in high variability of the measured RATE. Here, the secondary metrics (Recall and Precision) that are not affected, additionally assert the dominance of CaML over all baselines.

| | Split | # of Patients |
|---|---|---|
| Allopurinol | Test | 815,921 |
| Pregabalin | Test | 636,995 |
| Mirtazapine | Test | 623,980 |
| Indomethacin | Test | 560,380 |
| Colchicine | Test | 370,397 |
| Hydralazine | Test | 363,070 |
| Hydroxychloroquine | Test | 324,750 |
| Methotrexate | Test | 323,387 |
| Memantine | Test | 306,832 |
| Fentanyl | Test | 261,000 |
| Etodolac | Val | 438,854 |
| Azathioprine | Val | 100,000 |

Table 6: Held-out test and validation drugs for our single-drug meta-testing and meta-validation datasets for our Claims evaluation in Table 2. Drugs are unseen (excluded) during training. All drugs are known to cause pancytopenia [42]

| | Split | # of Patients |
|---|---|---|
| Allopurinol + Hydralazine | Test | 7,859 |
| Methotrexate + Hydroxychloroquine | Test | 25,716 |
| Pregabalin + Fentanyl | Test | 5,424 |
| Indomethacin + Colchicine | Test | 42,846 |
| Mirtazapine + Memantine | Test | 10,215 |

Table 7: Held-out test pairs of drugs for our meta-testing and meta-validation datasets in Appendix Table 5. Both drugs are unseen (excluded) during training. All drugs are known to cause pancytopenia [42]

| Split | # Perturbagens | # Cell-Lines | Mean #Cell Lines/Task |
|---|---|---|---|
| Meta-training | 9717 | 77 | 5.79 |
| Meta-validation | 304 | 11 | 9.99 |
| Meta-testing | 301 | 11 | 10.77 |

Table 8: Composition of the meta-training, meta-validation and meta-testing sets for the LINCS dataset. No cell lines or drugs (tasks) were shared across any of the splits.

| Hyperparameter | Search range |
|---|---|
| Num. of layers | $\{2, 4, 6\}$ |
| Dim. of hidden layers | $\{128, 256\}$ |
| Dropout | $\{0, 0.1\}$ |
| Learning rate | $\{3 \times 10^{-3}, 1 \times 10^{-3}, 3 \times 10^{-4}, 1 \times 10^{-4}\}$ |
| Meta learning rate | $\{1\}$ |
| Weight decay | $\{5 \times 10^{-3}\}$ |
| Reptile k | $\{1, 10, 50\}$ |
| L1 regularization coefficient | $\{0, 1 \times 10^{-7}, 5 \times 10^{-7}\}$ |

Table 9: Hyperparameter search space for **CaML** (our proposed method) on the medical claims dataset.

| Hyperparameter | Search range |
|---|---|
| Num. of como layers | $\{2, 4, 6\}$ |
| Num. of covariate layers | $\{2, 4, 6\}$ |
| Num. of propensity layers | $\{2, 4, 6\}$ |
| Num. of treatment layers | $\{2, 4, 6\}$ |
| Dim. of hidden como layers | $\{128, 256\}$ |
| Dim. of hidden covariate layers | $\{128, 256\}$ |
| Dim. of hidden treatment layers | $\{128, 256\}$ |
| Dim. of hidden propensity layers | $\{16, 32, 64, 128\}$ |
| Dropout | $\{0, 0.1\}$ |
| Learning rate | $\{3 \times 10^{-3}, 1 \times 10^{-3}, 3 \times 10^{-4}, 1 \times 10^{-4}\}$ |
| Meta learning rate | $\{1\}$ |
| Sin Weight decay | $\{0, 5 \times 10^{-3}\}$ |
| Pro Weight decay | $\{0, 5 \times 10^{-3}\}$ |
| GNN Weight decay | $\{0, 5 \times 10^{-3}\}$ |
| Reptile k | $\{1, 10, 50\}$ |
| L1 regularization coefficient | $\{0, 1 \times 10^{-7}, 5 \times 10^{-7}\}$ |

Table 10: Hyperparameter search space for **SIN** on the medical claims dataset. The SIN model consists of two stages, Stage 1 and Stage 2. For the Stage 1 model we searched the identical hyperparameter search space as for CaML (Table 9). For Stage 2, we used the hyperparameters shown in this table.

| Hyperparameter | Search range |
|---|---|
| Num. of covariate layers | $\{2, 4, 6\}$ |
| Num. of treatment layers | $\{2, 4, 6\}$ |
| Dim. of hidden treatment layers | $\{128, 256\}$ |
| Dim. of hidden covariate layers | $\{128, 256\}$ |
| Dropout | $\{0, 0.1\}$ |
| Independence regularization coefficient | $\{0, 0.01, 0.1, 1.0\}$ |
| Learning rate | $\{3 \times 10^{-3}, 1 \times 10^{-3}, 3 \times 10^{-4}, 1 \times 10^{-4}\}$ |
| Meta learning rate | $\{1\}$ |
| Weight decay | $\{5 \times 10^{-3}\}$ |
| Reptile k | $\{1, 10, 50\}$ |
| L1 regularization coefficient | $\{0, 1 \times 10^{-7}, 5 \times 10^{-7}\}$ |

Table 11: Hyperparameter search space for **GraphITE** on the medical claims dataset.

| Hyperparameter | Search range |
|---|---|
| Num. of out layers | $\{1, 2, 4\}$ |
| Num. of r layers | $\{2, 4, 6\}$ |
| Num. units p out | $\{32, 64, 128, 256\}$ |
| Num. units s out | $\{32, 64, 128, 256\}$ |
| Num. units s r | $\{32, 64, 128, 256\}$ |
| Num. units p r | $\{32, 64, 128, 256\}$ |
| Weight decay | $\{5 \times 10^{-3}\}$ |
| Orthogonal penalty | $\{0, 1 \times 10^{-5}, 1 \times 10^{-3}, 0.1\}$ |
| Private out | $\{\text{True, False }\}$ |
| Learning rate | $\{3 \times 10^{-3}, 1 \times 10^{-3}, 3 \times 10^{-4}, 1 \times 10^{-4}\}$ |

Table 12: Hyperparameter search space for **FlexTENet** on the medical claims dataset.

| Hyperparameter | Search range |
|---|---|
| Num. of out layers | $\{1, 2, 4\}$ |
| Num. of r layers | $\{2, 4, 6\}$ |
| Num. units out | $\{128, 256\}$ |
| Weight decay | $\{5 \times 10^{-3}\}$ |
| Penalty disc | $\{0, 1 \times 10^{-3}\}$ |
| Learning rate | $\{3 \times 10^{-3}, 1 \times 10^{-3}, 3 \times 10^{-4}, 1 \times 10^{-4}\}$ |

Table 13: Hyperparameter search space for **TARNet** on the medical claims dataset.

| Hyperparameter | Search range |
|---|---|
| Num. of out layers | $\{1, 2, 4\}$ |
| Num. of r layers | $\{2, 4, 6\}$ |
| Num. units r | $\{128, 256\}$ |
| Num. units out | $\{128, 256\}$ |
| Weight decay | $\{5 \times 10^{-3}\}$ |
| Learning rate | $\{3 \times 10^{-3}, 1 \times 10^{-3}, 3 \times 10^{-4}, 1 \times 10^{-4}\}$ |

Table 14: Hyperparameter search space for **DragonNet** on the medical claims dataset.

| Hyperparameter | Search range |
|---|---|
| Num. of estimators | $[50, 250]$ |
| Max depth | $[10, 50]$ |
| Min sample split | $[2, 8]$ |
| Criterion regress | $\{$squared error, absolute error$\}$ |
| Criterion binary | $\{$gini, entropy$\}$ |
| Max features | $\{$sqrt, log2, auto$\}$ |

Table 15: Hyperparameter search space for model-agnostic CATE estimators, i.e., **R-learner**, **X-learner**, **RA-learner**, and **T-learner** on the medical claims dataset.

| Hyperparameter | Search range |
|---|---|
| Num. of layers | $\{2, 4, 6\}$ |
| Dim. of hidden layers | $\{512, 1024\}$ |
| Dropout | $\{0, 0.1\}$ |
| Learning rate | $\{3 \times 10^{-3}, 1 \times 10^{-3}, 3 \times 10^{-4}, 1 \times 10^{-4}\}$ |
| Meta learning rate | $\{0.1, 0.5, 0.9\}$ |
| Weight decay | $\{0.1\}$ |
| Reptile k | $\{1, 2, 3\}$ |
| L1 regularization coefficient | $\{0, 1 \times 10^{-7}, 5 \times 10^{-7}\}$ |

Table 16: Hyperparameter search space for **CaML** (our proposed method) on the LINCS dataset.

| Hyperparameter | Search range |
|---|---|
| Num. of como layers | $\{2, 4, 6\}$ |
| Num. of covariates layers | $\{2, 4, 6\}$ |
| Num. of propensity layers | $\{2, 4, 6\}$ |
| Num. of treatment layers | $\{2, 4, 6\}$ |
| Dim. output | $\{128, 256\}$ |
| Dim. of hidden treatment layers | $\{128, 256\}$ |
| Dim. of hidden covariate layers | $\{128, 256\}$ |
| Dim. of hidden como layers | $\{128, 256\}$ |
| Dim. of hidden propensity layers | $\{16, 32, 64, 128\}$ |
| Model dim. | $\{512, 1024\}$ |
| Dropout | $\{0, 0.1\}$ |
| Learning rate | $\{3 \times 10^{-3}, 1 \times 10^{-3}, 3 \times 10^{-4}, 1 \times 10^{-4}\}$ |
| Meta learning rate | $\{0.1, 0.5, 0.9\}$ |
| Sin weight decay | $\{0.0, 0.005\}$ |
| Pro weight decay | $\{0.0, 0.005\}$ |
| GNN weight decay | $\{0.0, 0.005\}$ |
| Weight decay | $\{0.1\}$ |
| Reptile k | $\{1, 2, 3\}$ |
| L1 regularization coefficient | $\{0, 1 \times 10^{-7}, 5 \times 10^{-7}\}$ |

Table 17: Hyperparameter search space for the **SIN** baseline on the LINCS dataset.

| Hyperparameter | Search range |
|---|---|
| Num. of covariate layers | $\{2, 4, 6\}$ |
| Num. of treatment layers | $\{2, 4, 6\}$ |
| Num. of layers | $\{2, 4, 6\}$ |
| Dim. of hidden covariate layers | $\{128, 256\}$ |
| Independence regularization coefficient | $\{0, 0.01, 0.1, 1.0\}$ |
| Dropout | $\{0, 0.1\}$ |
| Model dim. | $\{512, 1024\}$ |
| Learning rate | $\{3 \times 10^{-3}, 1 \times 10^{-3}, 3 \times 10^{-4}, 1 \times 10^{-4}\}$ |
| Meta learning rate | $\{0.1, 0.5, 0.9\}$ |
| Weight decay | $\{0.1\}$ |
| Reptile k | $\{1, 2, 3\}$ |
| L1 regularization coefficient | $\{0, 1 \times 10^{-7}, 5 \times 10^{-7}\}$ |

Table 18: Hyperparameter search space for the **GraphITE** baseline on the LINCS dataset.

| Hyperparameter | Search range |
|---|---|
| Dropout | [1e-4,1e-1] |
| Learning rate | [1e-5,1e-3] |
| Weight decay | [1e-5,1e-2] |
| Adversarial temperature | [1,10] |
| Gamma | [0,30] |
| Num. of sampled neighbors | 0-10 |
| Dim. of hidden layers | $\{ 64, 128, 256, 512\}$ |

Table 19: The hyperparameter optimization search ranges used in the selection of the optimal model for the generation of knowledge graph node embeddings that would serve as intervention information for the medical claims dataset.

