# OpenReview forum: "Zero-shot causal learning"
_NeurIPS.cc/2023/Conference — NeurIPS 2023 spotlight_

### Official Review · Reviewer_fcYE · 2023-06-17

**Soundness:** 3 good
**Presentation:** 1 poor
**Contribution:** 2 fair
**Rating:** 7
**Confidence:** 4

**Summary:**

In this paper, authors have proposed causal meta-learning framework that trains a single model across several intervations and estimate CATE for novel interventions at inference time which were not present at the training time. They frame CATE estimation for each intervention as a separate meta-learning task. The proposed framework is evaulated on two real world large-scale datasets as medical claims and cell-line perturbations, which show the effectiveness of the proposed framework.

**Strengths:**

- A new meta-learning framework, called CaML, is proposed to estimate treatment effect for novel interventions by considering each intervention as a separate task. This is simple but interesting formulation.
- CaML is evaluated on two real world large-scale datasets as medical claims and cell-line perturbations, which show the effectiveness of the proposed framework
- Theoretical results are provided to bound the prediction error (I have not verified these results).
- Code is provided with the paper (I have not checked it).

**Weaknesses:**

- The idea seemed simple and interesting but I found it hard to follow, especially the results.

Why are the treatment effect estimates first calculated and used as kind of lables, and why not direclty learn the model? The selected CATE learner (RA-Learner) can create biased estimates.

There are typos which also make some sentences hand to interpret. Please revise accordingly.

In standard ML, use of validation dataset makes sense but what does validation tasks/interventions mean?

What do you mean by 'Given m outcomes of interest'? Does that mean the interventions have multiple outcomes discrete/continous?


**Questions:**

- What is the ratio of treated and untreated while preparing meta-datasets?
- CaML comes in different variations, like CaML - w/o RA-learner, etc -- if you could expalin all those at one place with other baselines then that would be easy to follow.
- Why very narrow range is considered for u in [0.999,0.998,0.995,0.99]?

**Limitations:**

Yes

---

> ### Author Rebuttal · Authors · 2023-08-09
>
> We thank the reviewer for their positive feedback on our problem formulation, meta-learning framework, real-world experimental evaluation, and theoretical results.
> We also appreciate the clarifying questions below, which have helped us to revise the manuscript to be more clear. If the reviewer has any further concerns, please let us know! If not, we would appreciate it if the reviewer would be willing to consider raising their score.
>
> > 1. The idea seemed simple and interesting but I found it hard to follow, especially the results.
>
> We thank the reviewer for this feedback and thoroughly revise the presentation for our camera-ready to clarify each of the points below.
>
> > 2. Why are the treatment effect estimates first calculated and used as kind of lables, and why not direclty learn the model? The selected CATE learner (RA-Learner) can create biased estimates.
>
> **TL;DR:** It is not possible to directly model CATE because it is unobserved. While there is an alternative to the pseudo-outcomes approach (i.e. modeling potential outcomes and then subtracting them), it has been widely shown to suffer from estimation bias (Künzel et al. 2019; Nie and Wager 2021). We discuss these points in Lines 90-93, Lines 133-138. We further empirically show via two baselines (S-learner and T-learners w/meta-learning; Tables 2 and 3) that this approach significantly underperforms compared to CaML.
>
>  In our revision, we will clarify and elaborate on the following points:
> - CATE estimation by modeling potential outcomes, and why it underperforms compared to the pseudo-outcome approach (due to regularization bias)
> - Why pseudo-outcome targets are unbiased, using an illustrative example of the RA-learner (see response to Reviewer o8uy’s point 1)
> - That CaML flexibly wraps *any* pseudo-outcome estimator and is not in any way restricted to the RA-learner
>
> *We have prepared a more detailed response for the question above, but cannot post it until the discussion period due to OpenReview character limitations.*
>
> > 3. There are typos which also make some sentences hand to interpret. Please revise accordingly.
>
> Thank you for pointing this out, and we apologize for the typos present in the initial submission. We have further thoroughly proofread the paper and made the necessary corrections to ensure clarity.
>
> > 4. What does validation tasks/interventions mean?
>
> In the CaML meta-learning framework, each task corresponds to CATE estimation for a single intervention (e.g., a single drug). Validation tasks contain interventions which were excluded from training, and allow us to validate our model’s ability to perform zero-shot CATE estimation on novel interventions. A validation task contains a natural experiment of many samples (e.g. patients) who received the unseen intervention, and many control samples who did not receive the intervention. All samples included in the validation task are excluded from the training.
>
> In our revised paper, we will elaborate on this point to make it more clear in the zero-shot evaluation subsection (Lines 236-244).
>
> > 5. What do you mean by 'Given m outcomes of interest'?
>
> In many settings, we may be interested in how an intervention will affect multiple outcomes. For example, in our LINCS evaluation, a single perturbation can affect many different genes. Our CATE estimation task thus corresponds to predicting the effect of a perturbation on the top 50 most differentially expressed genes $(Y_{1}, \dots ,Y_{50})$.
>
> We will include this concrete example on Line 182 to ensure that this point is clear.
>
> > 6. What is the ratio of treated and untreated while preparing meta-datasets?
>
> We use all available data, and hence this varies by the specific intervention and evaluation dataset. For instance, in our medical claims data provided to us by an insurance company, the number of patients that received each of the the held-out test drugs ranges from 261K to 815K patients, and the held-out control group is a sample of 1 million patients. Hence the fraction of treated patients for meta-testing tasks ranges from 20.7% to 44.9% depending on the test intervention. This class imbalance is an important reason for why we chose to use the RA-learner, as it is robust to class imbalance, as long as there is sufficient data in both classes to estimate the necessary nuisance models.
>
> We will include a supplementary table containing the treated vs. control ratios for all the tasks in our two real-world datasets.
>
> > 7. CaML comes in different variations, like CaML - w/o RA-learner, etc -- if you could explain all those at one place with other baselines then that would be easy to follow.
>
> While we had previously described these details in our supplements (Lines 968-1000), we will now present them in one place of the main paper.
>
> > 8. Why very narrow range is considered for $u$ in [0.999,0.998,0.995,0.99]?
>
> Our medical claims evaluation is tailored to the real-world deployment of CaML, in which we want to predict the effect of a drug on the likelihood of a patient developing the life-threatening side effect pancytopenia. Pancytopenia is a very rare event occurring in less than 0.3% of the patients in our dataset. Hence we choose this narrow range of u to isolate the small subset of high-risk patients from the vast majority of patients for whom there is no risk of pancytopenia onset. Notably, many approved drugs show life-threatening side effects (like pancytopenia) only very rarely. Hence, when considering a deployment scenario, it generally makes sense to consider narrow thresholds in order to focus on the small subset of patients at risk of a rare but serious side effect. In other settings, lower RATE thresholds may become more interesting: e.g., when considering treatments for which the majority of recipients have some treatment effect, such as surge pricing in online marketing.
>
> We elaborate on this rationale for choice of $u$ in our revision.

---

> > ### Comment · Reviewer_fcYE · 2023-08-10
> >
> > Thank you for your responses to my comments.
> >
> > In your rebuttal, you are using the term modelling 'potential outcomes' and 'pseudo-outcome' approach. It took me a while to understand the meaning of former. Don't you think, calling the former as 'Factual outcome' modelling will be more clear, especially when talking in context of pseudo-outcome modelling? If you talk only about potential outcome modelling then one can understand it from Rubin's PO framework. But 'potential outcomes' and 'pseudo-outcome' terms together creates confusion.
> >
> > Earlier, my only comment was on clarity of the paper. Authors have promised to revise the paper to add more clarity, as discussed above. So, I am assuming that will improve it and it will be easy to follow.
> >
> > I have one more query. I apologise, I did not ask it earlier. In this framework, which parts are relevant to causal literature. I mean, you created pseudo-outcomes using RA-Learner to create a dataset and then it becomes standard ML. So, as per my understanding, formulating the causal problem into zero-shot learning is the only novel part for the causal inference literature (and real-world application). Right? Because creating pseudo labels as well as the meta-learning part are existing techniques. Please clarify the novelty of the work to causal literature.
> >
> > I have also skimmed through comments of other reviewers and I did not find any major concern from other reviewers.

---

> > > ### Author Response · Authors · 2023-08-12
> > >
> > > Thank you for your thoughtful comments, feedback, and continued engagement with our paper!
> > >
> > > > In your rebuttal, you are using the term modelling 'potential outcomes' and 'pseudo-outcome' approach. It took me a while to understand the meaning of former. Don't you think, calling the former as 'Factual outcome' modelling will be more clear, especially when talking in context of pseudo-outcome modelling? If you talk only about potential outcome modelling then one can understand it from Rubin's PO framework. But 'potential outcomes' and 'pseudo-outcome' terms together creates confusion. Earlier, my only comment was on clarity of the paper. Authors have promised to revise the paper to add more clarity, as discussed above. So, I am assuming that will improve it and it will be easy to follow.
> > >
> > > We agree that the similarity between the terminology “potential outcomes” and “pseudo-outcome” may lead to confusion with readers. The problem with referring to modeling potential outcomes as  “factual outcome modeling” is that while the models are trained on factual data, they are then used to predict counterfactuals (i.e., we want to model Y(0) for treated patients, and Y(w) for control patients).
> > >
> > > In our revision, we will instead refer to the first approach as “modeling Y(w) and Y(0)”. We only refer to this approach in two places in the paper (Line 137, Line 259) as it is a mere baseline, and hence we believe it is OK to use this more verbose description to ensure clarity.
> > >
> > > > I have one more query. I apologise, I did not ask it earlier. In this framework, which parts are relevant to causal literature. (...) Please clarify the novelty of the work to causal literature. I have also skimmed through comments of other reviewers and I did not find any major concern from other reviewers.
> > >
> > > Thank you for raising this additional question. We take this as helpful feedback that we should more explicitly enumerate our contributions in our paper. We will revise our paper to do so, and to highlight the following contributions:
> > >
> > > **(1)**  Our first contribution is to formulate zero-shot CATE estimation. This problem is challenging (no training data for test interventions) and omnipresent: in many real-world settings, the overwhelming majority of possible interventions are unobserved. For instance, there are ~166.4 billion possible small molecules for drugs [1], whereas it is only possible to study a small number of drug candidates in clinical trials each year. Similarly, in online marketing, there is an infinite search space of possible advertisements, and a limited number of users to run A/B tests on. Achieving zero-shot CATE estimation will thus enable personalized treatment design across many domains.
> > >
> > > **(2)**  Second, our paper represents a milestone for the CATE methods literature by intentionally breaking evaluation conventions. Because treatment effects are unobserved, previous papers which introduced new CATE estimation methods typically relied on synthetic and semi-synthetic datasets [2, 3, 4]. However, this overreliance on simulated data has made it difficult to ascertain if there is actual progress that can benefit real-world applications [5]. In contrast, our study evaluates on two large-scale real-world datasets featuring millions of patients (Claims) and thousands of interventions (LINCS). The evaluation utilizes recent advancements: the emergence of biological cell perturbation data (LINCS) enabling observation of counterfactual ground truth by cloning a cell-line, and the RATE metric [6] which allows for evaluation on observational data (Claims).
> > >
> > > Our empirical evaluation yields several insights which we elaborate on in Section 6.3: (a) CaML’s zero-shot performance outperforms many prior methods trained directly on the test interventions (b) CaML is capable of higher-order generalization: trained on single interventions, it can can zero-shot generalize to pairs of (both novel) interventions (c) most methods besides CaML perform worse than simple zero-shot baselines (d) zero-shot CATE estimation can inform clinical decision-making, by identifying patients which are vulnerable to a life-threatening drug side effect (Claims eval).

---

> > > > ### Author Response · Authors · 2023-08-12
> > > > **continuation of response to contributions question**
> > > >
> > > > > pseudo labels as well as the meta-learning part are existing techniques
> > > >
> > > > **(3)** Successfully translating ideas from one ML subfield to another, and discovering that they unlock entirely new capabilities, is non-trivial and a contribution in its own right. For example, the Transformer architecture was originally introduced in 2017 for NLP. It wasn't until 2020 that its impact for computer vision was recognized with the landmark Vision Transformer (ViT) paper [7]. Similarly, progress in the subfield of Graph Neural Networks was sparked by the discovery that word embeddings (word2vec) can be extended to embed nodes in a graph (node2vec) [8].
> > > >
> > > > In the same spirit, developing a framework that allows for pseudo-outcome labels and meta-learning to be (successfully) applied to zero-shot CATE estimation requires careful design considerations. *This is because pseudo-outcome labels and meta-learning cannot be applied to zero-shot CATE estimation out-of-the-box.* Our key insight in designing the CaML framework is that, rather than training on the raw (W, X, Y) triplets, we can synthesize natural experiments from the data to construct a distinct CATE estimation task for each training intervention. This allows us to (a) unlock zero-shot capability for many recently developed pseudo-outcome estimators which were previously restricted to studying single interventions in isolation (b) leverage meta-learning algorithms (i.e. Reptile) for meta-model training, which requires task-structured data. We discover in our experiments that both of these factors contribute to CaML’s strong empirical performance and allows it to outperform all baselines and existing methods.
> > > >
> > > > **(4)** Finally, our theoretical contribution is to bound the zero-shot CATE prediction error as a function of several key parameters (e.g. how informative intervention information $w$ is of the true treatment effects). Compared to a standard generalization bound which usually has a $\sqrt{1/n}$ term, our main technical innovation involves bounding the variance by the smoothness of the function class plus Poincaré-type inequalities. When $\beta$ in Theorem 1 is much smaller than $1$ we achieve a tighter bound. This is unique to the causal inference setting in which we are learning a function $f = \Psi(w,x) \to \tau$ among a family $\mathcal{F}$, where $w$ is intervention information (e.g. a drug’s attributes) and $x$ is individual features (e.g. patient medical history), and $f$ is assumed to be smooth with respect to $w$.
> > > >
> > > > **References**
> > > > [1] Ruddigkeit, Lars, et al. "Enumeration of 166 billion organic small molecules in the chemical universe database GDB-17." Journal of chemical information and modeling 52.11 (2012): 2864-2875.
> > > > [2] Curth, Alicia, and Mihaela van der Schaar. "Nonparametric estimation of heterogeneous treatment effects: From theory to learning algorithms." AISTATS. (2021).
> > > > [3] Nie, Xinkun, and Stefan Wager. "Quasi-oracle estimation of heterogeneous treatment effects." Biometrika (2021).
> > > > [4] Kennedy, Edward H. "Towards optimal doubly robust estimation of heterogeneous causal effects." arXiv (2020).
> > > > [5] Curth, Alicia, et al. "Really doing great at estimating CATE? a critical look at ML benchmarking practices in treatment effect estimation." NeurIPS. (2021).
> > > > [6]  Yadlowsky, Steve, et al. "Evaluating treatment prioritization rules via rank-weighted average treatment effects." arXiv preprint arXiv:2111.07966 (2021).
> > > > [7] Dosovitskiy, Alexey, et al. "An image is worth 16x16 words: Transformers for image recognition at scale." ICLR (2021).
> > > > [8] Grover, Aditya, and Jure Leskovec. "node2vec: Scalable feature learning for networks." KDD. (2016).

---

> > > > > ### Comment · Reviewer_fcYE · 2023-08-14
> > > > >
> > > > > Thanks for the clarifications. I agree formulation of zero-shot CATE estimation and appliction of ideas from other fields is not trivial, and have more than sufficient novelty for acceptance. Just make it clear in the paper.
> > > > >
> > > > > Overall, authors clarified some points, promised to revise some points and I don't see any major concern from fellow reviewers. So, I raise my evaluation by one point.

---

### Official Review · Reviewer_o8uy · 2023-06-28

**Soundness:** 3 good
**Presentation:** 3 good
**Contribution:** 3 good
**Rating:** 7
**Confidence:** 3

**Summary:**

The paper proposes a zero short cate estimation algorithm. Building natural experiments sub datasets and estimating pseudo outcomes the paper attempts at estimating the cate of unseen interventions

**Strengths:**

- interesting premise
- interesting and novel method
- adequate review of related works
- good experimentation and discussion


**Weaknesses:**

- unclear why the pseudo outcome target \tau is unbiased
- unclear how the algorithm would fair if the training data was biased in the prevalence of specific interventions . For example if the available data only contains a small subset of possible interventions it is possible that the generalisation of the model is not adequate to span the entire set of possible interventions

**Questions:**

- if the authors could address the latter point in the weakness section

**Limitations:**

- adequate discussion of limitations and impact

---

> ### Author Rebuttal · Authors · 2023-08-09
>
> > The paper proposes a zero short cate estimation algorithm. Building natural experiments sub datasets and estimating pseudo outcomes the paper attempts at estimating the cate of unseen interventions
> > - interesting premise
> > - interesting and novel method
> > - adequate review of related works
> > - good experimentation and discussion
>
> We thank the reviewer for their overwhelmingly positive feedback on our problem formulation, methodology, experimental evaluation, and discussion.
>
> > 1. unclear why the pseudo outcome target $\tau$ is unbiased
>
> Thank you for pointing out that this could be more clear. As we note in Line 170, here unbiased means that the expected value of the pseudo outcome target $\tilde{\tau}$ is equal to $\tau$, i.e., $E[\tilde{\tau} |X= x] = \tau(x)$.
>
> We will revise our paper to more explicitly refer to the prior work which develops these pseudo outcome targets and analyzes their theoretical properties, under standard causal inference assumptions of unconfoundedness, consistency, and overlap (Lines 97-100) [1, 2, 3, 4]. While each pseudo-outcome estimation method is unique, performance varies by problem setting, and hence CaML flexibly wraps any pseudo-outcome label.
>
> Prior work shows that pseudo-outcomes are unbiased, when they are constructed for a given sample using correctly specified nuisance models trained on an independent split of the data (e.g., via cross-fitting or treated/control split). To further clarify this point, we revise our paper to include an illustrative example for the RA-learner [1] and show that its pseudo-outcome labels are unbiased CATE estimates, i.e.:
> \begin{align*}
> \mathbb{E}\left[\tilde{\tau} | X = x\right] = \tau(x) = \mathbb{E}\left[Y(1) - Y(0)|X=x\right]
> \end{align*}
>
> The pseudo-outcome $\tilde{\tau}$ for the RA-learner is defined as $
>     \tilde{\tau}_i = Y_i - \hat{\mu}_0(X_i)$
> for treated units ($W_i=1$), and
> $
>     \tilde{\tau}_i = \hat{\mu}_1(X_i) - Y_i
> $
> for control units ($W_i=0$). Here, $\hat{\mu}_0(X), \hat{\mu}_1(X)$ denote unbiased and correctly specified nuisance models for the outcomes $Y(0)$ and $Y(1)$ respectively. In other words, $\mathbb{E}\left[\hat{\mu}_0(x)\right] = \mu_0(x) = \mathbb{E}\left[Y(0) | X = x\right]$ and $\mathbb{E}\left[\hat{\mu}_1(x)\right] = \mu_1(x) = \mathbb{E}\left[Y(1) | X = x\right]$.
>
> We consider the treated and control units separately. For treated units ($W_i=1$), we have:
> \begin{align*}
>     \tilde{\tau}_i = Y_i - \hat{\mu}_0(X_i).
> \end{align*}
>
> Hence, their expectation, conditioned on covariates $X$ can be written as:
>
> \begin{align*}
>     \mathbb{E}\left[\tilde{\tau} | X=x\right] = \mathbb{E}\left[Y - \hat{\mu}_0(X) | X=x \right] = \mathbb{E}\left[Y|X=x\right] - \mathbb{E}\left[\hat{\mu}_0(X) |X=x\right]
> \end{align*}
> \begin{align*}
>  = \mathbb{E}\left[Y|X=x\right] - \mathbb{E}\left[Y(0)|X=x\right] = \tau(x),
> \end{align*}
>
> which by applying the consistency assumption (paper Line 98) for treated units is equivelant to:
>
> \begin{align*}
>      \mathbb{E}\left[Y(1)|X=x\right] - \mathbb{E}\left[Y(0)|X=x\right] = \mathbb{E}\left[Y(1) - Y(0)|X=x\right] = \tau(x).
> \end{align*}
>
> Analogously, we can make the same argument for control units ($W_i=0$). Here, the pseudo-outcome is computed as:
> \begin{align*}
>     \tilde{\tau}_i = \hat{\mu}_1(X_i) - Y_i.
> \end{align*}
>
> Hence, we have
> \begin{align*} \mathbb{E}\left[\tilde{\tau} | X=x\right] = \mathbb{E}\left[\hat{\mu}_1(X) - Y | X=x \right] = \mathbb{E}\left[\hat{\mu}_1(X)|X=x \right]- \mathbb{E}\left[Y|X=x\right],
> \end{align*}
>
> which under consistency (for control units) is equivalent to:
>
> \begin{align*}
>      \mathbb{E}\left[Y(1)|X=x\right] - \mathbb{E}\left[Y(0)|X=x\right] = \mathbb{E}\left[Y(1) - Y(0)|X=x\right] = \tau(x).
> \end{align*}
>
> **References**
>
> [1] Curth, Alicia, and Mihaela van der Schaar. "Nonparametric estimation of heterogeneous treatment effects: From theory to learning algorithms." International Conference on Artificial Intelligence and Statistics. PMLR, 2021.
> [2] Kennedy, Edward H. "Towards optimal doubly robust estimation of heterogeneous causal effects." arXiv preprint arXiv:2004.14497 (2020).
> [3] Künzel, Sören R., et al. "Metalearners for estimating heterogeneous treatment effects using machine learning." Proceedings of the national academy of sciences 116.10 (2019): 4156-4165.
> [4] Nie, Xinkun, and Stefan Wager. "Quasi-oracle estimation of heterogeneous treatment effects." Biometrika 108.2 (2021): 299-319.
>
> > 2. unclear how the algorithm would fair if the training data was biased in the prevalence of specific interventions . For example if the available data only contains a small subset of possible interventions...
>
> We agree that this point should be further clarified, and will revise our paper to do so.
>
> In general, limited training examples, or biases in the training data, will degrade model performance—and the CaML algorithm is no exception in this regard. For instance, if there are too few examples of prior interventions (e.g. only a handful of training drugs), then zero-shot generalization may become more challenging. Therefore, we agree that it is important to study the robustness of CaML’s performance to limitations in the training dataset.
>
> To clarify this point, we have implemented additional experiments in which we downsample the number of training interventions. In the **PDF attached to our global response**, we present two result figures for our cell-line perturbations evaluation. Experiments for the (considerably larger) medical claims dataset are still running and will be included in the camera-ready.
>
> As expected, we find that: (1) zero-shot capabilities improve as the set of unique training interventions increases in size and (2) we can still achieve strong performance on smaller datasets (e.g. runs with 60% and 80%, of the interventions achieve similar performance). We will include these new experiments in our revised paper.

---

> > ### Comment · Reviewer_o8uy · 2023-08-11
> >
> > I have read the authors rebuttal and I will maintain my score of acceptance.
> > I appreciate the detailed response on the subject of bias in \tau and the bias of the datasets. Looking forward to the updated paper with the supplementary experiments

---

### Official Review · Reviewer_qej5 · 2023-07-05

**Soundness:** 3 good
**Presentation:** 2 fair
**Contribution:** 3 good
**Rating:** 7
**Confidence:** 4

**Summary:**

The paper presents a novel framework for zero-shot causal learning, which is the problem of predicting the personalized effects of a novel intervention that has no historical data. The framework, called CaML (Causal Meta-learning), formulates the prediction of each intervention's effect as a meta-learning task, and trains a single meta-model across thousands of tasks, each constructed by sampling an intervention. The paper demonstrates the effectiveness of CaML on two real-world datasets in large-scale medical claims and cell-line perturbations, and shows that CaML can outperform existing methods that are either trained on the test interventions or capable of zero-shot generalization. The paper also provides a theoretical analysis of CaML's zero-shot generalization bound. The paper contributes to the field of causal inference by unlocking zero-shot capability for many recently developed CATE estimation methods that were previously restricted to single interventions.


**Strengths:**

The paper presents an original approach to addressing the challenging problem of zero-shot causal learning, which has important applications in various domains including personalized medicine, public policy, and online marketing. The paper's framework is comprehensive and rigorous, and its effectiveness is demonstrated through experiments on two real-world datasets in large-scale medical claims and cell-line perturbations. Additionally, the paper unlocks the zero-shot capability for many recently developed CATE estimation methods, which were previously restricted to single interventions, and provides a promising direction for future research in causal inference.

**Weaknesses:**

While the paper provides a thorough evaluation of its framework using metrics such as RATE and PEHE, future research should explore additional evaluation methods and metrics that can provide a more comprehensive understanding of the performance of zero-shot causal learning methods. Additionally, the paper's section on societal impacts could benefit from a more detailed discussion of the potential risks and limitations of the proposed approach. For instance, the risk of underrepresentation of underserved communities in the training data could lead to biased results, which could ultimately have negative societal impacts. Therefore, future research should address these limitations and consider approaches to mitigate potential risks associated with the proposed framework.

**Questions:**

Above weaknesses part

**Limitations:**

Above weaknesses part

---

> ### Author Rebuttal · Authors · 2023-08-09
>
> > (...) The paper presents an original approach to addressing the challenging problem of zero-shot causal learning, which has important applications in various domains including personalized medicine, public policy, and online marketing. The paper's framework is comprehensive and rigorous, and its effectiveness is demonstrated through experiments on two real-world datasets in large-scale medical claims and cell-line perturbations. Additionally, the paper unlocks the zero-shot capability for many recently developed CATE estimation methods, which were previously restricted to single interventions, and provides a promising direction for future research in causal inference.
>
> We thank the reviewer for their overwhelmingly positive feedback on the CaML framework, our experimental analyses, theoretical results, and the broader impacts of zero-shot CATE estimation.
>
> > 1. While the paper provides a thorough evaluation of its framework using metrics such as RATE and PEHE, future research should explore additional evaluation methods and metrics that can provide a more comprehensive understanding of the performance of zero-shot causal learning methods.
>
> We agree that an important direction for future work is to explore additional evaluation metrics. In this paper, we focus on PEHE and RATE as they are prominent metrics of CATE error (PEHE being prominent in the seminal literature [1, 2, 3], and RATE in more recent work [4, 5, 6]). We have extended our discussion to stress the importance of developing and exploring further metrics for CATE estimation, including measures of calibration and bias (a point we further address below in our discussion of the risks of CATE models).
>
> **References**
>
> [1] Hill, Jennifer L. "Bayesian nonparametric modeling for causal inference." Journal of Computational and Graphical Statistics 20.1 (2011): 217-240.
> [2] Shalit, Uri, Fredrik D. Johansson, and David Sontag. "Estimating individual treatment effect: generalization bounds and algorithms." International conference on machine learning. PMLR, 2017.
> [3] Künzel, Sören R., et al. "Metalearners for estimating heterogeneous treatment effects using machine learning." Proceedings of the national academy of sciences 116.10 (2019): 4156-4165.
> [4] Offer-Westort, Molly, Leah R. Rosenzweig, and Susan Athey. "Battling the CoronavirusInfodemic'Among Social Media Users in Africa." arXiv preprint arXiv:2212.13638 (2022).
> [5] Metz-Peeters, Maike. The Effects of Mandatory Speed Limits on Crash Frequency-A Causal Machine Learning Approach. No. 982. Ruhr Economic Papers, 2023.
> [6] Xu, Yizhe, et al. "Principled estimation and evaluation of treatment effect heterogeneity: A case study application to dabigatran for patients with atrial fibrillation." Journal of Biomedical Informatics (2023): 104420.
>
> > 2. Additionally, the paper's section on societal impacts could benefit from a more detailed discussion of the potential risks and limitations of the proposed approach. For instance, the risk of underrepresentation of underserved communities in the training data could lead to biased results, which could ultimately have negative societal impacts. Therefore, future research should address these limitations and consider approaches to mitigate potential risks associated with the proposed framework.
>
> Indeed, it is of utmost importance to quantify, understand, and mitigate the effects of personalized causal models on underserved communities—and more broadly, to understand and address the risks of deploying CATE estimators in practice.
>
> Regarding equitable and fair CATE estimation, two promising directions are process-oriented and outcome-oriented approaches [1]. A process-oriented approach would quantify model error (e.g. calibration, false positive rates) for underserved groups to understand whether they may be disparately impacted by the deployment of CATE models [2, 3]. An outcome-oriented approach would quantify the expected impact of the deployment of such models on utility of each demographic subgroup [4, 5], and tailor resulting decision policies to increase the utility of the underserved population. Both are important areas which we will investigate in our follow-up work.
>
> Regarding the broader risks of deploying CATE models, there is a potential for overreliance on CATE models, leading to decreased human oversight and decision-making. While CATE estimators can provide valuable insights into treatment effects, they should not replace clinical judgment and expertise. Relying solely on model predictions without considering individual patient circumstances and medical expertise could lead to inappropriate or ineffective treatments. Additionally, the deployment of CATE models could raise privacy concerns. These models typically require access to individual patient data to estimate personalized treatment effects accurately. Ensuring the privacy and security of this sensitive information is crucial to avoid potential data breaches or unauthorized access, which could harm patients and erode public trust in healthcare systems.
>
> In our camera-ready, we will include a significantly extended discussion on these key points.
>
> **References**
>
> [1] Corbett-Davies, Sam, et al. "The measure and mismeasure of fairness." arXiv preprint arXiv:1808.00023 (2023).
> [2] Obermeyer, Ziad, et al. "Dissecting racial bias in an algorithm used to manage the health of populations." Science 366.6464 (2019): 447-453.
> [3] Seyyed-Kalantari, Laleh, et al. "Underdiagnosis bias of artificial intelligence algorithms applied to chest radiographs in under-served patient populations." Nature medicine 27.12 (2021): 2176-2182.
> [4] Nilforoshan, Hamed, et al. "Causal conceptions of fairness and their consequences." International Conference on Machine Learning. PMLR, 2022.
> [5] Liu, Lydia T., et al. "Delayed impact of fair machine learning." International Conference on Machine Learning. PMLR, 2018.

---

> > ### Comment · Reviewer_uxCL · 2023-08-18
> > **Changing score**
> >
> > Updated my score based on the response and discussions

---

### Official Review · Reviewer_uxCL · 2023-07-06

**Soundness:** 2 fair
**Presentation:** 3 good
**Contribution:** 3 good
**Rating:** 7
**Confidence:** 3

**Summary:**

In this paper, the authors propose a new Meta Learning algorithm called CaML - to predict the personalized effects of a novel intervention. The idea is neat, and its applicability seems reasonable given the empirical results.

**Strengths:**

- The paper's algorithm seems to perform well for novel settings.
- The PAC-like theorem is a nice addition

Based on the response I have upgraded my score to Accept

**Weaknesses:**

- I'd like to see a theorem like the one in https://arxiv.org/pdf/2112.07602.pdf (Lemma 1 or Lemma 2), which would validate unbiasedness in different settings (assuming that pre-treatment covariates are used)
- A more complex set of theorems (which requires extensions that are more complex) would be ones where the distribution of the actual meta-datasets are drifting. I think even then strong results can be obtained, which would allow the use of CaML even in settings where the novel interventions are drawn from different distributions. These might not be feasible for this paper but maybe worth addressing

**Questions:**

- Are there any other guarantees you can make on the out of sample error without a theorem like Lemma 1 above?
- What covariates can be used?

**Limitations:**

The authors address some limitations - I have outlined others in the "Weaknesses" section above

---

> ### Author Rebuttal · Authors · 2023-08-09
>
> > (...) The idea is neat, and its applicability seems reasonable given the empirical results (...) The paper's algorithm seems to perform well for novel settings (...) The PAC-like theorem is a nice addition
>
> We thank reviewer uxCL for their positive feedback on our method, the strength of our empirical results, and our zero-shot generalization bound. We prove the additional theorem requested by the reviewer below, which validates the unbiasedness of our training objective.
>
>
> If the reviewer has any further concerns, please let us know! If not, we would appreciate it if the reviewer would be willing to consider raising their score.
>
> > 1. I'd like to see a theorem (...) which would validate unbiasedness in different settings (assuming that pre-treatment covariates are used)
>
> We thank the reviewer for this insightful comment. We have updated our paper to prove a similar lemma as Lemma 1 from Tripuraneni et al. [1], which we agree strengthens our arguments. We have also cited Tripuraneni et al. as our proof follows a similar structure.
>
> For each task (corresponding to a single intervention) we obtain pseudo-outcome labels (denoted as $\tilde{\tau}$), which are unbiased but noisy estimates of CATE (denoted as $\tau$). Specifically, we have $\mathbb{E}[\tilde{\tau} | X=x] = \tau(x)$ for a sample with pre-treatment covariates $x$. An extensive literature develops methods to obtain these pseudo-outcomes [2, 3, 4, 5]. CaML flexibly wraps any of these existing methods to obtain labels for each task.
>
> We now prove that by leveraging these unbiased (yet noisy) pseudo-outcome labels, we can formulate an unbiased (yet noisy) optimization objective to train and evaluate an estimator for the true CATE. In other words, if estimator A outperforms estimator B on the noisy pseudo-outcome loss (in expectation), then it also holds that estimator A will outperform estimator B at predicting the true CATE (in expectation).
>
> More formally, we consider parametrized CATE estimators $\Psi_{\theta} \colon \mathbb{R}^e \times \mathbb{R}^d \to \mathbb{R}$ that take as input intervention information $w \in \mathbb{R}^e$ (e.g., a drug's attributes) and individual features $x \in \mathbb{R}^d$ (e.g., patient medical history) to return a scalar for the estimated CATE (e.g., the effect of the drug on patient health).
>
> We denote the loss function $L$ with regard to a CATE estimator $\Psi$ and a target $y$ as
>
> $L(\Psi,y) = \left(\Psi(w, x) - y\right)^2$
>
> As in Theorem 1 in our paper, we assume pseudo-outcome targets $\tilde{\tau}$ during training satisfy $\tilde{\tau}=\tau+\xi$ where $\tau$ is the true (unobserved) CATE and $\xi$ is an independent zero-mean noise.
>
> **Lemma 10**: Given two different CATE estimators $\hat{\Psi_{\theta_{1}}}$, $\hat{\Psi_{\theta_{2}}}$, parameterized by $\theta_{1}$ and $\theta_{2}$:
>
> $\mathbb{E}[L(\hat{\Psi_{\theta_{1}}}, \tilde{\tau})] \leq \mathbb{E}[L(\hat{\Psi_{\theta_{2}}}, \tilde{\tau})] \implies
> \mathbb{E}[L(\hat{\Psi_{\theta_{1}}}, \tau)] \leq \mathbb{E}[L(\hat{\Psi_{\theta_{2}}}, \tau)]$
>
> The proof is provided in the PDF shared with the AC on OpenReview (as external links in reviewer responses are disallowed by NeurIPS).
>
> > Are there any other guarantees you can make on the out of sample error without a theorem like Lemma 1 above? (...)  A more complex set of theorems (...) would be ones where the distribution of the actual meta-datasets are drifting. (...) These might not be feasible for this paper but maybe worth addressing.  (...)
>
> We thank the reviewer for highlighting these important questions for theoretical inquiry. As we are the first paper to extensively explore whether zero-shot CATE estimation is possible in real-world settings, we focused primarily on its empirical aspects; we develop the CaML framework and demonstrate its strong performance at generalizing to novel interventions across two real-world datasets.
>
> We recognize the exciting theoretical questions our work raises, and in future research we will indeed delve deeper into addressing these exact questions. To study the scenario where testing interventions are drawn from a different distribution from the training interventions, we need additional assumptions on the nature of the domain shift to provide non-trivial OOD generalization guarantees. In the simplest case where we have linear models, if the test intervention is in the span of training interventions, we can prove that CATE for the test intervention is learnable. Alternatively, in the case where the test intervention distribution has sufficient overlap with training distributions, we can use generalization bounds like the ones in Ben-David et al. [6] or Zhang et al. [7] to bound the test error using training error plus a term bounding the divergence between two distributions. Thank you for your valuable feedback, which will shape our considerations for subsequent extensions of our paper, which we are actively investigating,
>
> > 3. What covariates can be used.
>
> Consistent with the CATE estimation literature, covariates must be pre-treatment. For example, in our Claims evaluation we use patient medical history. We will clarify this in our revised paper.
>
> **References**
>
> [1] Tripuraneni, et al., 2021. Meta-Analysis of Randomized Experiments with Applications to Heavy-Tailed Response Data.
> [2] Künzel, et al., 2019. Metalearners for estimating heterogeneous treatment effects using machine learning.
> [3] Kennedy, et al., 2020. Towards optimal doubly robust estimation of heterogeneous causal effects.
> [4] Wager, et al., 2018. Estimation and inference of heterogeneous treatment effects using random forests.
> [5] Curth, et al., 2021. Nonparametric estimation of heterogeneous treatment effects: From theory to learning algorithms.
> [6] Ben-David, et al., 2010. A theory of learning from different domains.
> [7] Zhang, et al., 2019. Bridging theory and algorithm for domain adaptation.

---

### Author Rebuttal · Authors · 2023-08-09

We thank the reviewers for their valuable feedback, recommendations for improvements, and generally positive reviews of our work. Overall, it appears that the reviewers found the problem setting of zero-shot CATE estimation important, well-motivated, and applicable (Reviewers qej5, o8uy, uxCL), noted the novelty of our algorithm and its strong empirical performance in our experiments (Reviewers fcYE, o8uy, qej5, uxCL), and appreciated our theoretical contribution of a finite-sample bound on the zero-shot error (Reviewers uxCL, qej5, fcYE).


The reviewers raise several questions, which we appreciate and answer in our individual responses. To address Reviewer uxCL's request for additional theoretical guarantees, we prove a new lemma (see **note below** for proof) which validates that by wrapping unbiased pseudo-outcome labels from prior work, we can formulate an unbiased optimization objective to train and evaluate an estimator for the true CATE. We were additionally inspired by Reviewer o8uy's question concerning zero-shot generalization under limited training data, and conducted new experiments which show how CaML’s performance varies when the number of training interventions is limited (see **PDF attached** to this response).

These insightful comments and suggestions have helped us to enhance the clarity and strength of our manuscript. Thank you!

***

**Note**: Per NeurIPS OpenReview policy, we can't include text (besides figure captions) in the attached PDF. To adhere to this, we've uploaded the lemma proof separately and shared an anonymous link with the AC, following NeurIPS instructions for code requests (copied below). Please inform us during the discussion period if there are any issues in accessing the proof, and if an alternative sharing method is required.

> If you were asked by the reviewers to provide code, please send an anonymized link to the AC in a separate comment (make sure the code itself and all related files and file names are also completely anonymized).

---

### Decision · Program_Chairs · 2023-09-21

**Decision:**

Accept (spotlight)

**Comment:**

This paper proposes a meta-learning framework to estimate causal effects for novel treatments (zero-shot). The method treats each intervention+data-specific task to predict pseudo outcomes or treatment effects, and a meta-learner that samples an intervention to predict the pseudo outcomes. Theoretical results in the form of a generalization bound on zero-shot learner are presented. Experiments demonstrate utility in two tasks.

Guaranteeing good zero-shot performance for causal effect estimation is a highly challenging task. Positivity assumptions are required for any causal effect estimation. Therefore in my reading of the paper some subtle inherent smoothness assumptions might have been made that are a necessity to allow such extrapolation to novel interventions. It may be that the meta-learning process learns to estimate this structure easily compared to other vanilla methods. However, a nuanced discussion of this does not appear in the manuscript. The authors demonstrate the utility of their method in for predicting drug side effects and gene expression tasks. It is unclear how much the method generalizes beyond these tasks. I encourage the authors to include a discussion on this in the final version of the paper.

All reviewers have acknowledged the utility and novelty of the contribution. Authors have successfully addressed questions around potential deterioration when fewer training interventions are available that may jeopardize generalization. Authors provide additional empirical evaluation to test this deterioration in their experiments.

Considering the utility of the empirical contributions, I recommend an accept.